# Kupffer cells prevent pancreatic ductal adenocarcinoma metastasis to the liver in mice

Stacy K. Thomas [1,2], Max M. Wattenberg [1,2], Shaanti Choi-Bose[1,2], Mark Uhlik[3,5], Ben Harrison[3], Heather Coho[1,2], Christopher R. Cassella[1,2], Meredith L. Stone[1,2], Dhruv Patel[1,2], Kelly Markowitz [1,2], Devora Delman[1,2], Michael Chisamore[4], Jeremy Drees[3], Nandita Bose[3] & Gregory L. Beatty[1,2] ✉

Although macrophages contribute to cancer cell dissemination, immune evasion, and metastatic outgrowth, they have also been reported to coordinate tumor-specific immune responses. We therefore hypothesized that macrophage polarization could be modulated therapeutically to prevent metastasis. Here, we show that macrophages respond to β-glucan (odetiglucan) treatment by inhibiting liver metastasis. β-glucan activated liver-resident macrophages (Kupffer cells), suppressed cancer cell proliferation, and invoked productive T cell-mediated responses against liver metastasis in pancreatic cancer mouse models. Although excluded from metastatic lesions, Kupffer cells were critical for the anti-metastatic activity of β-glucan, which also required T cells. Furthermore, β-glucan drove T cell activation and macrophage re-polarization in liver metastases in mice and humans and sensitized metastatic lesions to anti-PD1 therapy. These findings demonstrate the significance of macrophage function in metastasis and identify Kupffer cells as a potential therapeutic target against pancreatic cancer metastasis to the liver.

Metastasis is among the leading causes of cancer-associated deaths[1]. Metastatic lesions form from tumor cells that disseminate from a primary tumor and seed a distant tissue, such as the liver[2]. During this process, disseminated tumor cells (DTCs) must evade elimination by local immune cells and adapt to a new microenvironment. Once acclimated, the DTC proliferates to establish a metastatic colony which ultimately manifests as an expanding metastatic lesion[2]. In pancreatic ductal adenocarcinoma (PDAC), this metastatic cascade is often already ongoing at the time of diagnosis with nearly 50% of patients presenting with distant metastasis[3–5]. In PDAC, the most common site of metastasis is the liver, with the lung and peritoneum being slightly less common[5]. Notably, nearly 80% of patients with PDAC will develop liver metastasis, and the 5-year survival rate for metastatic PDAC is a mere 3%, despite standard-of-care chemotherapy[6]. Thus, treatments to successfully intervene on the metastatic cascade in PDAC are a clinical unmet need[3].

For many solid malignancies, therapeutic strategies designed to harness the host immune system have produced clinical benefits. However, immunotherapy, such as immune checkpoint blockade (ICB), has not yet demonstrated reproducible benefits for patients with PDAC[7–9]. Immune resistance in PDAC is thought to be due, in part, to a profoundly immunosuppressive tumor microenvironment (TME) enriched in myeloid cells with a scarcity of effector T cells[10–12]. However, in other solid cancers, such as melanoma and lung cancer, liver metastasis also associates with reduced efficacy of ICB[13–17]. Consistent with this clinical observation, preclinical models suggest that liver macrophages are engendered with immune suppressive features that inhibit productive T cell surveillance in cancer[15].

[1]Division of Hematology-Oncology, Department of Medicine, Perelman School of Medicine, University of Pennsylvania, Philadelphia, PA, USA. [2]Abramson Cancer Center, Perelman School of Medicine, University of Pennsylvania, Philadelphia, PA, USA. [3]HiberCell Inc, Roseville, MN, USA. [4]Merck & Co., Inc., Kenilworth, NJ, USA. [5]Present address: OncXerna, Waltham, MA, USA. ✉e-mail: gregory.beatty@pennmedicine.upenn.edu

One approach to overcoming myeloid immunosuppression in cancer involves the use of drugs designed to deplete macrophages[18,19]. However, an alternative approach is to redirect the phenotype of macrophages using drugs that exploit the plasticity of macrophage biology and the potential of macrophages to mediate anti-tumor activity[20–24]. For instance, anti-tumor functions of macrophages include phagocytosis of tumor cells, production of secreted molecules like cytokines and reactive oxygen species (ROS) that are tumoricidal or cytostatic, and antigen presentation to activate tumor-specific T cells[20]. Ultimately, factors present within the surrounding microenvironment shape the phenotype and functions of macrophages.

Here, we investigated the interaction between DTCs and macrophages within the liver microenvironment. To redirect liver macrophage biology, we studied β-glucans (BG), which are molecules found on the cell surface of yeast and other single-celled organisms and that are known to engender macrophages with anti-tumor properties. β-glucans are a pathogen-associated molecular pattern (PAMP) and signal through the Dectin-1 receptor on macrophages[25,26]. Stimulation with a β-glucan polarizes macrophages towards an anti-tumor phenotype, enhances phagocytic potential, and can induce trained immunity[27–32]. Here, we studied a soluble β−1,3/1,6 glucan (odetiglucan) that is already in clinical development. Our data show that a soluble β−1,3/1,6 glucan is an immunomodulator of the liver metastatic process. β-glucan treatment redirected liver macrophages from pro- to anti-metastatic. Specifically, Kupffer cells (KCs) were found to coordinate anti-metastatic activity and cooperate with T cells to restrain liver metastasis. Further, β-glucan treatment combined with ICB (anti-PD1) to produce a survival benefit after metastatic challenge in mouse models and altered the myeloid and T cell response in metastatic lesions from patients with triple-negative breast cancer and advanced melanoma. Thus, our studies not only identify a soluble β−1,3/1,6 glucan as a therapeutic approach for leveraging anti-metastatic immunity but also reveal KCs as a key cellular target for regulating immunosurveillance and sensitizing metastatic lesions to ICB.

## Results

### Macrophages differentially regulate early and late phases of liver metastasis

To study the role of macrophages in the seeding (early) and outgrowth (late) phases of the metastatic cascade in the liver (Supplementary Fig. 1a), we first assessed the interaction of macrophages with tumor cells in metastatic liver lesions from humans (Supplementary Fig. 1b, c) and mice (Supplementary Fig. 1d, e). Macrophages (human: CD68⁺, mouse: F4/80⁺) were identified within three distinct regions of the liver: (i) metastatic lesion, (ii) periphery of a metastatic lesion and (iii) normal adjacent liver. Macrophages tended to be less prominent in metastatic lesions compared to adjacent liver tissue in mice, but this trend was not seen in humans. To examine cell-cell interactions between liver macrophages and tumor cells as they seed and colonize the liver, we next modeled liver metastasis via intraportal (iPo) injection of tumor cells (Supplementary Fig. 1f). We purposely chose this approach because metastasis is an early event in PDAC[3–5] and with this model, we could study the late stages of the metastatic cascade when disseminated tumors cells encounter a distant organ. While some seeding tumor cells were observed to interact with liver macrophages, the majority were not in contact with a macrophage (Supplementary Fig. 1g–i), suggesting that macrophages in the liver may not routinely survey for tumor cells during metastatic seeding.

We next sought to investigate cell-cell interactions between tumor cells and distinct liver macrophage subsets. We performed scRNAseq on normal mouse livers and identified *Clec4f* as a differentially expressed gene that distinguished liver resident Kupffer cells (KCs) from bone marrow derived macrophages (BMDMs) (Supplementary Fig. 2a–d), consistent with prior literature[33]. Based on this finding, we used Clec4f as a KC-specific marker to study interactions

between tumor cells and liver macrophage subsets using multiplex immunohistochemistry (mIHC). Seeding tumor cells were more frequently detected in contact with a F4/80⁺ BMDM than F4/80⁺Clec4f⁺ KC (Supplementary Fig. 2e–h). Further, in established metastases (outgrowth phase), BMDMs largely existed within lesions. In contrast and as expected, KCs, which reside in the liver sinusoids, were restricted to the adjacent liver tissue (Supplementary Fig. 2i–l). Thus, ontologically distinct macrophage subsets reside in unique spatially defined regions associated with metastatic lesions and may differentially engage with tumors during the metastatic process.

We next assessed the role of liver macrophages during metastatic seeding. To deplete macrophages, we used clodronate encapsulated liposomes (CEL)[34] administered prior to metastatic challenge (Supplementary Fig. 3a). CEL depleted both KCs and BMDMs in the liver (Supplementary Fig. 3b–d). However, macrophage depletion produced no impact on metastatic seeding (Supplementary Fig. 3e, f, Supplementary Fig. 4). We also tested the specific role of KCs in metastatic seeding. To do this, diphtheria toxin (DT) was administered to Clec4f-cre-tdTomato⁺/⁻Rosa26-LSL-DTR⁺/⁻ (Clec4f^DTR) mice[35] (Supplementary Fig. 5a–e). Similarly, KC depletion produced no impact on metastatic seeding (Supplementary Fig. 5f–h). Thus, metastatic seeding in the liver can occur independently of macrophages.

Macrophages have previously been identified as supportive of metastasis in the liver[36]. To test a role for macrophages during the outgrowth phase of metastasis, we administered CEL to deplete liver macrophages after metastatic seeding (Supplementary Fig. 6a). In this setting, macrophage depletion produced a significant decrease in tumor burden in the liver (Supplementary Fig. 6b–e). We next tested the role of KCs in regulating metastatic outgrowth. To do this, Clec4f^DTR mice were treated daily with DT after tumor seeding to deplete KCs (Supplementary Fig. 7a). In this model, metastatic burden was not affected by KC depletion (Supplementary Fig. 7b–d). Taken together, these data are consistent with a role for tumor-associated macrophages, rather than Kupffer cells, in supporting metastatic outgrowth as has been previously proposed[36].

### β-glucan treatment inhibits liver metastasis

Macrophage biology is pliable and determined by signals received from the surrounding microenvironment[21–24]. To this end, external stimuli can engender macrophages with either pro-tumor or anti-tumor activity (Fig. 1a). For instance, β-glucan (BG), a pathogen-associated molecular pattern (PAMP), acts as a myeloid agonist to enhance the anti-tumor functions of macrophages (e.g. by increased phagocytosis of tumor cells)[30–32,37,38]. Myeloid agonists have been extensively studied in the context of primary tumor lesions; however, strategies to harness macrophages for anti-metastatic activity remain ill-defined[30–32]. We assessed the capacity to redirect liver macrophages with anti-metastatic activity by testing the impact of a soluble BG (odetiglucan) on liver metastasis in mice (Fig. 1b, c). Odetiglucan is currently in clinical development and is a Dectin-1 (CLEC7A) agonist. Tumor burden was decreased in the livers of BG-treated mice compared to control mice in multiple models (Fig. 1d–g, Supplementary Fig. 8a–c). BG also inhibited tumor cell proliferation, as measured by Ki67 expression (Fig. 1h, Supplementary Fig. 8d). Histological examination of metastatic lesions from BG-treated mice showed decreased CK19⁺ tumor cell density, suggesting that treatment induced a contraction in the tumor cell population (Fig. 1i, j, Supplementary Fig. 8e). We considered the possibility that the anti-metastatic activity of BG may not be restricted to the liver. Regarding this, we found that BG produced a significant reduction in tumor burden in a model of PDAC lung metastasis (Supplementary Fig. 9). This finding is consistent with prior work showing that whole BG particles can suppress lung metastasis in mouse models[39]. However, the anti-metastatic activity of BG has not been previously reported in PDAC. Therefore, we focused our subsequent investigations on the

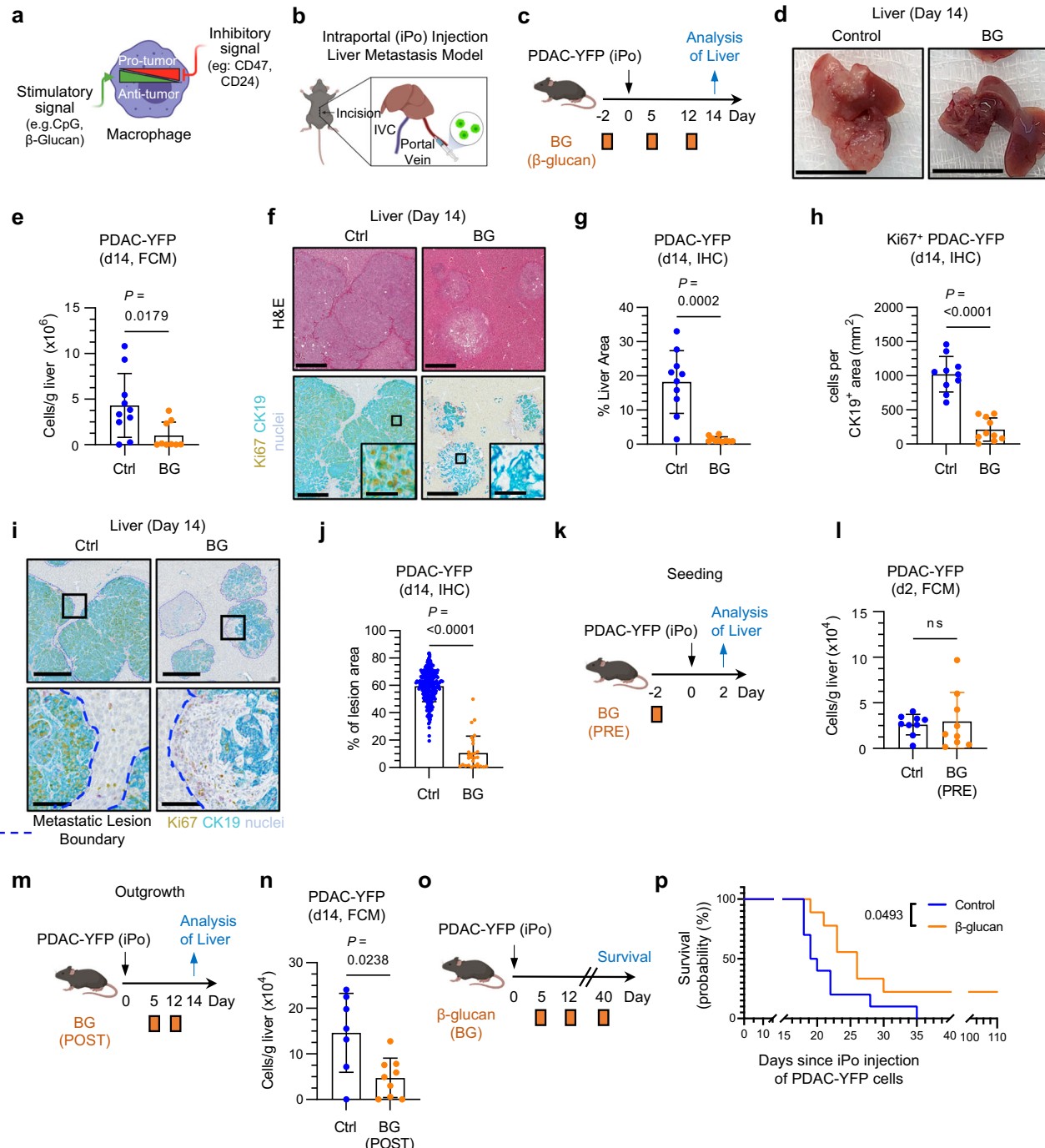

**Fig. 1 | β-glucan treatment restricts liver metastasis. a** Schematic of macrophage plasticity. **b** Schematic of Intraportal (iPo) injection model. **c** Study design for (**d**–**j**). Data are representative of 4 independent experiments. **d** Representative gross images of dissected livers. Scale bar, 1 cm. **e** Quantification of PDAC-YFP tumor cells detected in the liver by FCM in control (n = 10) and BG-treated (n = 9) mice. **f** Representative images of livers from control and BG-treated mice after iPo injection of PDAC-YFP cells. Tissues stained using hematoxylin and eosin (H&E) (top) and mIHC (bottom). Scale bar, 500 μm. Insets show a metastatic lesion. Scale bar, 100 μm. **g** PDAC-YFP (CK19+) tumor cells detected in the liver by IHC and shown as a percent of the liver area in control and BG-treated mice (n = 10 each). **h** CK19+Ki67+ PDAC-YFP tumor cells detected in liver by IHC and shown as cells per CK19+ area (mm²) in control and BG-treated mice (n = 10 each). **i** Representative images of liver stained by mIHC as indicated in (**f**). Metastatic lesions, blue dashed

lines. Top row images, scale bar, 500 μm. Bottom row images. Scale bar, 100 μm. **j** PDAC-YFP (CK19+) tumor cells detected in liver metastatic lesions by IHC and shown as percent of metastatic lesion area as drawn in **i**. Control (n = 259) and BG-treated (n = 28) tumor lesions were analyzed from control and BG-treated livers (n = 10 each). **k** Study design for (**l**). Data are representative of 4 independent experiments. **l** Quantification of PDAC-YFP tumor cells detected in the liver on Day 2 by FCM in control (n = 9) and BG-treated (n = 9) mice. **m** Study design for **n**. Data are representative of 3 independent experiments. **n** Quantification of PDAC-YFP tumor cells detected in the liver by FCM in control (n = 7) and BG (n = 9) treated mice. **o** Study design for (**p**). Kaplan–Meier plot for control (n = 10) and BG (n = 9) treated mice. Data are representative of 2 independent experiments. Unpaired two-tailed Welch's t-test (**e, g, h, j, l, n**) and Log-rank test (**p**). Mean ± SD is shown (**e, f, g, h, j, l, n**). FCM, flow cytometry; IHC, immunohistochemistry; iPo, intraportal.

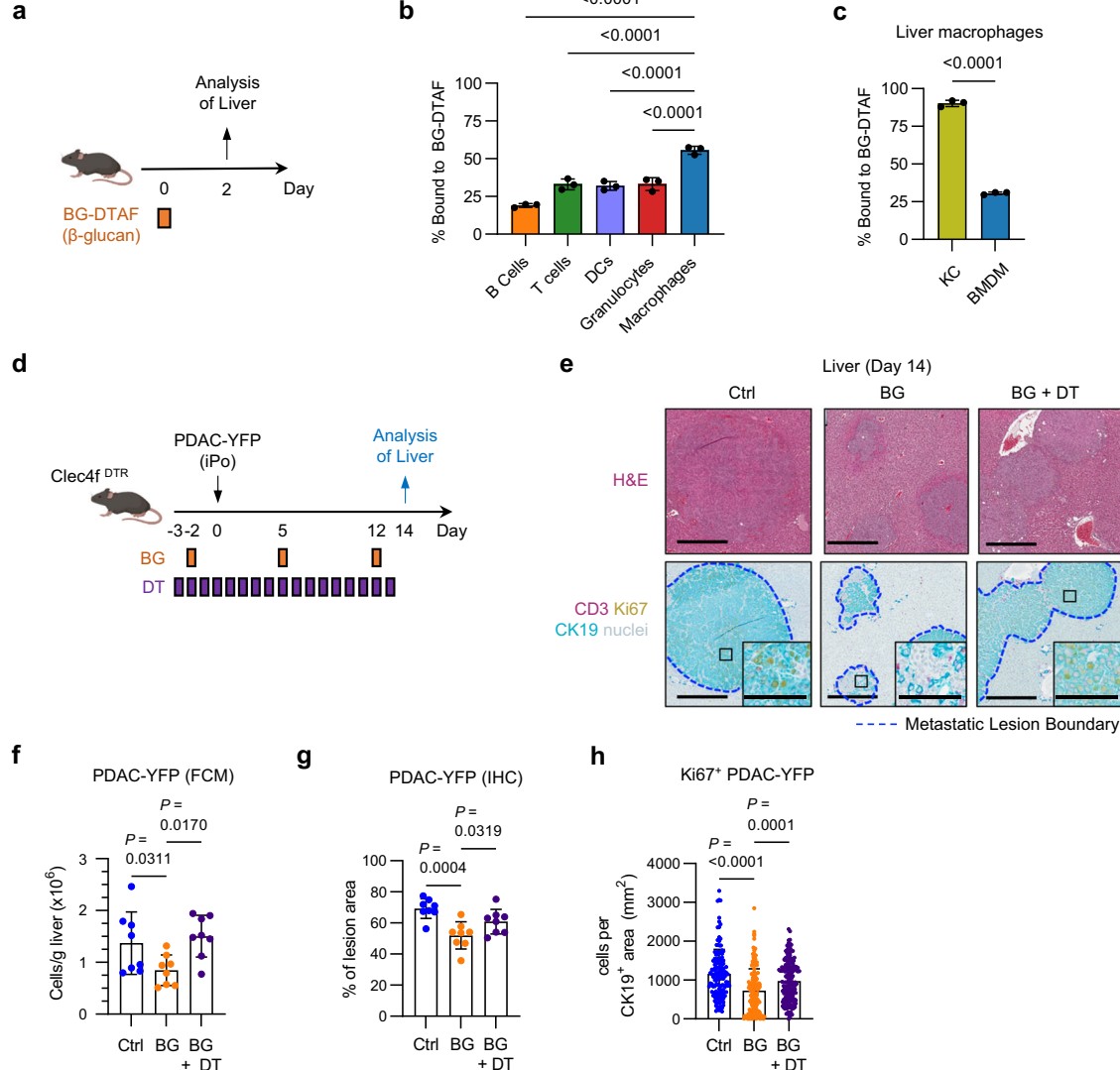

**Fig. 2 | Kupffer cells are necessary for the liver anti-metastatic effects of BG.**
**a** Study design for (**b**). Healthy mice (n = 3) were injected with 5-(4,6-Dichloro-triazinyl) Aminofluorescein (DTAF) labeled β-glucan. Two hours later, mice were euthanized, and livers analyzed. **b** Frequency of each liver immune subsets bound to β-glucan-DTAF, as measured by FCM. **c** Frequency of KC and BMDMs bound to β-glucan-DTAF, as measured by FCM in n = 3 healthy mice. **d** Study design for (**e–h**). **e** Representative images of livers from control, BG and BG + DT treated mice 14 days after iPo injection of PDAC-YFP cells. Tissues stained using hematoxylin and eosin (H&E) (top) and mIHC (bottom) to detect CK19 (teal), CD3 (purple), Ki67 (yellow), and nuclei (blue, hematoxylin). Scale bar, 500 μm. Insets show a meta-static lesion. Scale bar, 100μm. **f** Quantification of PDAC-YFP tumor cells detected in the liver on Day 14 by FCM (n = 8 mice/grp). **g** PDAC-YFP (CK19⁺) tumor cells

detected in liver metastatic lesions on Day 14 by IHC and shown as percent of metastatic lesion area as drawn in (**e**). (n = 8 mice/group). **h** Proliferating PDAC-YFP (CK19⁺Ki67⁺) tumor cells detected in liver metastatic lesions on Day 14 by IHC and shown as percent of metastatic lesion area as drawn in (**i**). Control (n = 138), BG-treated (n = 172) and BG + DT treated (n = 175) individual tumor lesions were analyzed from n = 8 livers per group. Data are representative of 2 independent experiments. Unpaired two-tailed Welch's t test (**c**) and one-way ANOVA with Sidak's test (**b, f, g, h**). Mean ± SD is shown (**b, c, f, g, h**). BG-DTAF, β-glucan-5-(4,6-Dichlorotriazinyl) Aminofluorescein; DCs, dendritic cells; KC, Kupffer cells; BMDMs, bone marrow-derived macrophages; iPo, intraportal; DT, diphtheria toxin; BG, β-glucan.

anti-metastatic mechanisms of BG in the PDAC liver metastasis setting.

To understand the mechanisms underlying the anti-metastatic activity of BG against liver metastases, we next examined the impact of BG on the early (seeding) and late (outgrowth) phases of liver metastasis. BG treatment produced no impact on metastatic seeding, consistent with our prior findings showing that the seeding phase of metastasis occurs independent of macrophages (Fig. 1k, l, Supplementary Fig. 10). In contrast, BG treatment, when initiated after metastatic seeding, inhibited liver metastasis (Fig. 1m, n). This finding is consistent with the role for BG in restricting the outgrowth phase of metastasis. Based on this finding, we tested the timing of BG administration for conditioning the liver with an anti-metastatic phenotype.

To do this, BG was delivered before (pre), after (post), or both before and after metastatic challenge (Supplementary Fig. 11). Regardless of the schedule of BG treatment, mice treated with BG showed significantly lower tumor burden in the liver compared to control mice. Moreover, BG treatment prolonged overall survival after metastatic challenge (Fig. 1o, p). Taken together, these data show that the liver can be conditioned with anti-metastatic activity via treatment with BG.

**Anti-metastatic activity triggered by BG requires Kupffer cells**
To understand the cellular mechanism of BG-induced anti-metastatic activity, we next investigated BG binding to leukocyte populations in the liver. To do this, DTAF was conjugated to BG as a fluorescent tag to monitor its biodistribution in vivo (Fig. 2a). At 2 hours after

administration, BG was detected bound to a proportion of all major leukocyte subsets, including T cells and dendritic cells (DCs) (Fig. 2b). However, macrophages had the highest frequency (55%) of cells bound with BG. Among liver macrophage subsets, most KCs were found to bind BG (90%), whereas a minority (30%) of BMDMs bound BG in vivo (Fig. 2c). Based on this finding, we next tested the role of macrophages in mediating the anti-metastatic activity of BG. To do this, we first depleted macrophages using CEL (Supplementary Fig. 12a). As expected, CEL treatment, when administered with BG, caused a reduction in total liver macrophages including KCs, whereas depletion of tumor-associated macrophages was modest (Supplementary Fig. 12b–e), In addition, CEL treatment inhibited the anti-metastatic activity of BG (Supplementary Fig. 12f–k). Based on the differential depletion effects of CEL on tumor-associated macrophages and KCs, we next specifically considered the role of KCs in the anti-metastatic activity of BG. To assess the role for KCs in the anti-metastatic activity induced by BG, we used the Clec4f$^{DTR}$ mouse model and administered DT daily beginning prior to BG treatment to deplete KCs (Fig. 2d). KC depletion inhibited the anti-metastatic activity of BG (Fig. 2e–h). Taken together, these findings show that macrophages, specifically KCs, are required for BG-induced anti-metastatic activity in the liver.

BG is known to bind to receptors on leukocytes and in doing so, cause cell activation. Dectin-1 (Clec7a) is a pattern recognition receptor (PRR) that recognizes BG and is expressed on macrophages[40]. In the liver, Dectin-1 was expressed by both KCs and BMDMs, and treatment with BG produced a decrease in Dectin-1 surface expression (Supplementary Fig. 13), consistent with internalization of Dectin-1 following receptor engagement by BG[41]. To interrogate changes in macrophage biology triggered by treatment with BG, we analyzed the transcriptome of liver macrophages at 48 hours after BG treatment in non-tumor bearing mice using scRNAseq (Fig. 3a). To do this, we sequenced liver cells from untreated (n = 4) and BG-treated (n = 4) mice. We then used unbiased clustering and a dimensionality reduction through uniform manifold approximation and projection (UMAP) to define liver macrophage subsets (Supplementary Fig. 14a, b). This approach revealed two major subsets of macrophages (KCs and BMDMs) defined based on Clec4f expression (Fig. 3b). Treatment with BG (compared to control) triggered an upregulation of several differentially expressed genes in KCs (12 genes) and BMDMs (80 genes) with 8 genes shared between the KC and BMDM subsets (Fig. 3c, d; Supplementary Fig. 14c). Based on the requirement of KCs for BG anti-metastatic activity, we next focused on transcriptional changes observed in KCs in response to BG treatment. Gene set enrichment analysis revealed an enrichment in genes associated with inflammatory response, interferon gamma response and interferon alpha response (Fig. 3e). Consistent with this, KCs showed increased expression of IFN-response genes (e.g. Irf7, Stat1, Isg15), MHC-related genes (e.g. B2m, Tap1), and genes associated with antigen presentation (APC-related genes; e.g. Fcgr1) (Fig. 3f–i; Supplementary Fig. 14d, e). These changes suggest that BG triggers activation of liver macrophages and a shift in their phenotype. Consistent with this, CD206, a marker associated with pro-tumor macrophages, was found to significantly decrease on KCs in non-tumor bearing mice in response to BG while there was increased expression of MHCI (H2-K$^b$/H2-D$^b$) on KCs (Fig. 3j)[42]. Similarly, in the setting of liver metastasis, BG treatment selectively caused a decrease in CD206 expression on KCs. In addition, both KCs and BMDMs responded to BG with an increase in CD38 expression, a marker associated with an anti-tumor phenotype (Fig. 3k). Thus, BG triggers biological changes in liver macrophages, including KCs and BMDMs, consistent with their potential to mediate anti-tumor activity.

## KCs cooperate with T cells to mediate BG-induced anti-metastatic activity

Macrophages can exert anti-tumor effects through several mechanisms including phagocytosis of tumor cells, tumor cell killing through secretion of ROS or reactive nitrogen species (RNS), release of cytokines and chemokines, and serving as antigen-presenting cells to activate T cells. Since transcriptional profiling revealed that KCs respond to BG via upregulation of genes associated with antigen presentation and interferon response (Fig. 3), we next investigated the interaction of liver macrophages with T cells in the liver. In normal livers, T cells were more frequently detected in contact with a KC than a BMDM (Fig. 4a–c). Further, BG treatment increased the frequency of T cells in contact with a KC, whereas fewer BMDMs were observed in contact with a T cell after BG treatment. Based on this finding, we next examined transcriptional changes in T cells at 48 hours after BG treatment using scRNAseq (Supplementary Fig 14). Notably, T and NK cells in the liver were found to have the highest number of DEGs after BG treatment compared to control (Supplementary Fig 14f). Among T and NK cells, a population of Cxcr6$^+$ Cd4$^+$ T cells showed the greatest increase in DEGs (363 genes) with upregulated DEGs (n = 267) associated with IFN-related (e.g. Irf7, Isg15, Oas3) and cell cycle pathways (e.g. Pcna, E2f1, Cdk4), and downregulated DEGs (n = 96) associated with cell differentiation (e.g. Tcf7, Cd27) and cell migration (e.g. Ccr7, S1pr1) (Fig. 4d–j). These findings suggest that KCs may trigger an anti-metastatic adaptive immune response in the liver.

Next, we examined T cell infiltration in the metastatic liver. Although the overall density of T cells in the liver was not affected by BG treatment (Supplementary Fig. 15a), there was an increase in CD3$^+$ T cell infiltration and in proliferating CD3$^+$ Ki67$^+$ T cells in metastatic lesions of BG-treated mice compared to control mice (Fig. 5a, b). In addition, T cell infiltration within metastatic lesions coincided with areas of decreased CK19$^+$ tumor cell density. We termed these histologically defined regions as "contraction zones" and the regions with a denser CK19$^+$ cell population as "tumor zones" (Fig. 5c). Contraction zones contained an increased density of T cells compared to tumor zones, suggesting a potential role for T cells in the anti-metastatic activity induced by BG (Fig. 5d, e). Thus, we next depleted T cells using anti-CD4 and anti-CD8 depleting antibodies prior to and during BG treatment (Fig. 5f). T cell depletion produced a significant reduction in T cells in the blood and liver (Supplementary Fig. 15b–d). Notably, T cell depletion produced an increase in metastatic burden in the livers of BG-treated mice (Fig. 5g–i, Supplementary Fig. 15e, f). Moreover, T cell depletion was associated with an increase in CK19$^+$ density in metastatic lesions (Fig. 5j). Finally, depletion of KCs in Clec4f$^{DTR}$ mice using DT prevented the formation of T cell infiltrated contraction zones and the corresponding decrease in tumor zones triggered by BG (Fig. 5k–m). Together, these data show that BG targeting of KCs is required for the formation of T cell infiltrated "contraction zones" and that T cells are critical to anti-metastatic potential.

## Kupffer cells and T cells coordinate BG-induced activation of BMDMs

We next investigated the mechanism by which KCs and T cells coordinate anti-metastatic activity. T cells can participate in immunosurveillance by acting as cytotoxic effectors or by secreting soluble factors to recruit a productive inflammatory response. Regarding this, BG showed a major early impact (day 2) on CD4$^+$ T cells (Fig. 4e), which can play a coordinator role in cancer immunosurveillance, driving recruitment and anti-tumor programming of myeloid cells[43]. Therefore, we hypothesized that acute activation of KCs and T cells might coordinate an ongoing innate immune response to control metastasis. Consistent with this, scRNAseq analysis of cells isolated from livers of mice 14 days after metastatic challenge showed non-KC myeloid cells, and not T/NK cells or KCs, to have the greatest number of DEGs after BG compared to control (Fig. 6a–c). Clustering of non-KC myeloid cells revealed five transcriptionally unique BMDM subsets and two conventional dendritic cell subsets (Fig. 6d). Of these macrophage and dendritic cell populations, DEG analysis nominated

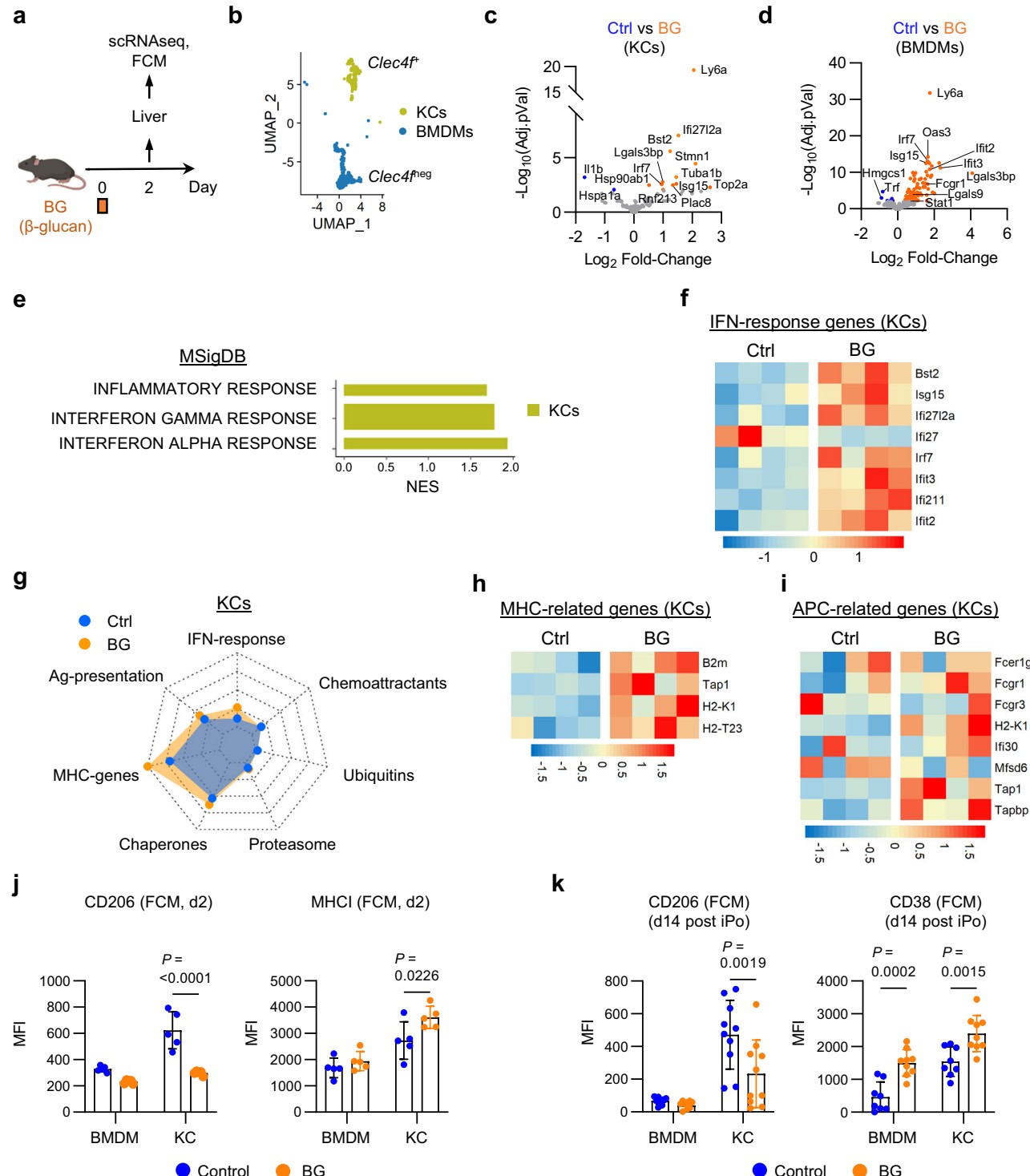

**Fig. 3 | β-glucan upregulates antigen presentation pathways in Kupffer cells.**
**a** Study design for (**b**–**i**). **b** UMAP of macrophage populations identified from
scRNAseq of control (n = 4) and BG-treated (n = 4) mouse livers. KCs (green) and
BMDMs (blue) are defined by *Clec4f* expression. **c**, **d** Volcano plots of differentially
expressed genes by pseudobulk analysis among Kupffer cells (**c**) and BMDMs (**d**) in
control versus BG-treated livers. Annotated upregulated (orange) and down-
regulated (blue) genes after BG are highlighted. Significance determined by log2-
FoldChange < −0.5 or > 0.5 and adjusted *p-value* < 0.05. **e** Bar graph of selected
gene sets identified as enriched by GSEA in KCs after treatment with BG. **f** Heatmap
of IFN-response genes upregulated in KCs after BG. Scale bars indicate log₂[TPM]
scaled by row across samples. **g** Radar plot showing selected gene sets expressed
by KCs in control (blue) and BG (orange) treated livers. Axis represents the mean
TPM of selected gene set associated genes with range 0 to 172 TPM. **h**–**i** Heatmaps

of MHC-related (**h**) and Antigen-presenting cell-related (**i**) genes upregulated in KCs
after BG. Scale bars indicate log₂[TPM] scaled by row across samples.
**j** Quantification of CD206 (left) and MHC class I (H2-Kb/H2-Db, right) expression by
BMDMs and KCs in the liver on Day 2 by FCM (n = 5 mice/group). **k** Quantification of
CD206 (left) and CD38 (right) expression by BMDMs and KCs in the liver on Day 14
by FCM (n = 10 mice/group). Two-way ANOVA with correction for multiple com-
parison (**j** and **k**). Mean ± SD is shown (**j** and **k**). Flow cytometry data are repre-
sentative of 2 independent experiments. BMDM, bone marrow derived
macrophage; FCM, flow cytometry; iPo, intraportal; KC, Kupffer cell; scRNAseq,
single cell RNA sequencing; Ctrl, control; IFN, interferon; MHC, major histo-
compatibility complex; APC, antigen presenting cell; NES, normalized enrichment
score; MFI, mean fluorescence intensity.

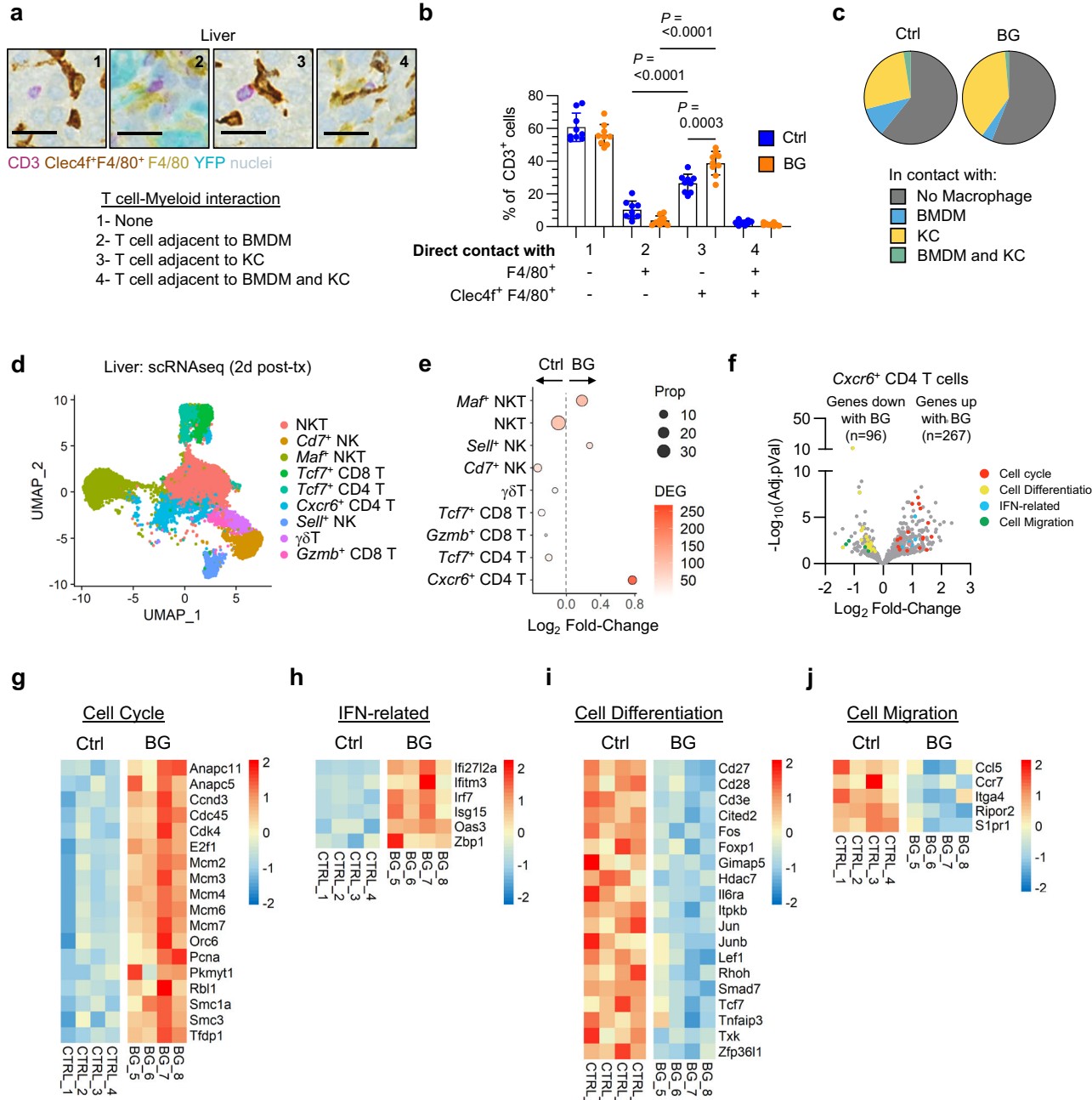

**Fig. 4 | β-glucan treatment drives Kupffer cell – T cell interactions.**
**a** Representative images of livers from mice at 14 day after iPo injection of PDAC-YFP tumor cells. Tissues stained using mIHC to detect YFP⁺ tumor cells (teal), CD3 (purple), F4/80 (yellow), Clec4f (brown), and nuclei (blue, hematoxylin). Images show T cells in contact with (1) no macrophage, (2) a BMDM, (3) a KC and (4) both a BMDM and a KC. Scale bar, 25μm. **b** Quantification of cell-cell interactions between T cells (CD3⁺) and liver macrophage subsets (Clec4f⁺F4/80⁺, KCs; F4/80⁺, BMDMs) in mouse livers of control and BG-treated mice at 14 days after iPo injection of 200,000 PDAC-YFP tumor cells (n = 9 mice/grp). One-way ANOVA with Tukey's test. Mean ± SD is shown. **c** Pie charts summarizing data shown in (**b**). **d** UMAP of T cell/NK populations identified by scRNAseq in control and BG-treated livers on day 2. **e** Dot plot of Log₂ Fold-Change in T cell/NK population frequency in BG-treated

livers compared to control livers. Size represents the proportion of total T cell/NK and color indicates the number of differentially expressed genes in BG versus control. **f**, Volcano plot of differentially expressed genes by pseudobulk analysis among *Cxcr6* CD4⁺ T cells in control versus BG-treated livers. Annotations highlight genes associated with the cell cycle (red), cell differentiation (yellow), IFN (blue) and cell migration (green). Significance determined by log2FoldChange < −0.5 or > 0.5 and adjusted *p* value < 0.05. **g**–**j** Heatmaps of cell cycle (**g**), IFN-related (**h**), cell differentiation (**i**) and cell migration (**j**) gene set associated genes, differentially expressed in *Cxcr6* CD4⁺ T cells from control and BG-treated livers on day 2. Scale bars indicate log₂[TPM] scaled by row across samples. BMDM, bone marrow-derived macrophage; KC, Kupffer cell; scRNAseq, single-cell RNA sequencing; Ctrl, control; IFN, interferon.

*Spp1*⁺MΦs (*Spp1*⁺BMDMs) as a key responder to BG at day 14 (Fig. 6e). Notably, *Spp1* expressing macrophages are commonly found in human liver metastasis, such as in colorectal cancer[44]. Furthermore, consistent with a role for BG in shifting Spp1⁺ MΦs from a pro-tumor to an anti-tumor phenotype, gene set enrichment analysis of *Spp1*⁺

MΦs revealed upregulation of genes related to antigen presentation, IL-1, immune regulatory pathways (e.g., *Arg1*, *Cd274*) and nitric oxide production (e.g., *Nos2*, *Tnf*, *Sod2*) (Fig. 6f–j). Given this, we next turned to define the requirement for KCs and T cells in mediating anti-tumor programming of *Spp1*⁺MΦs. As Nos2 (iNOS), a hallmark of

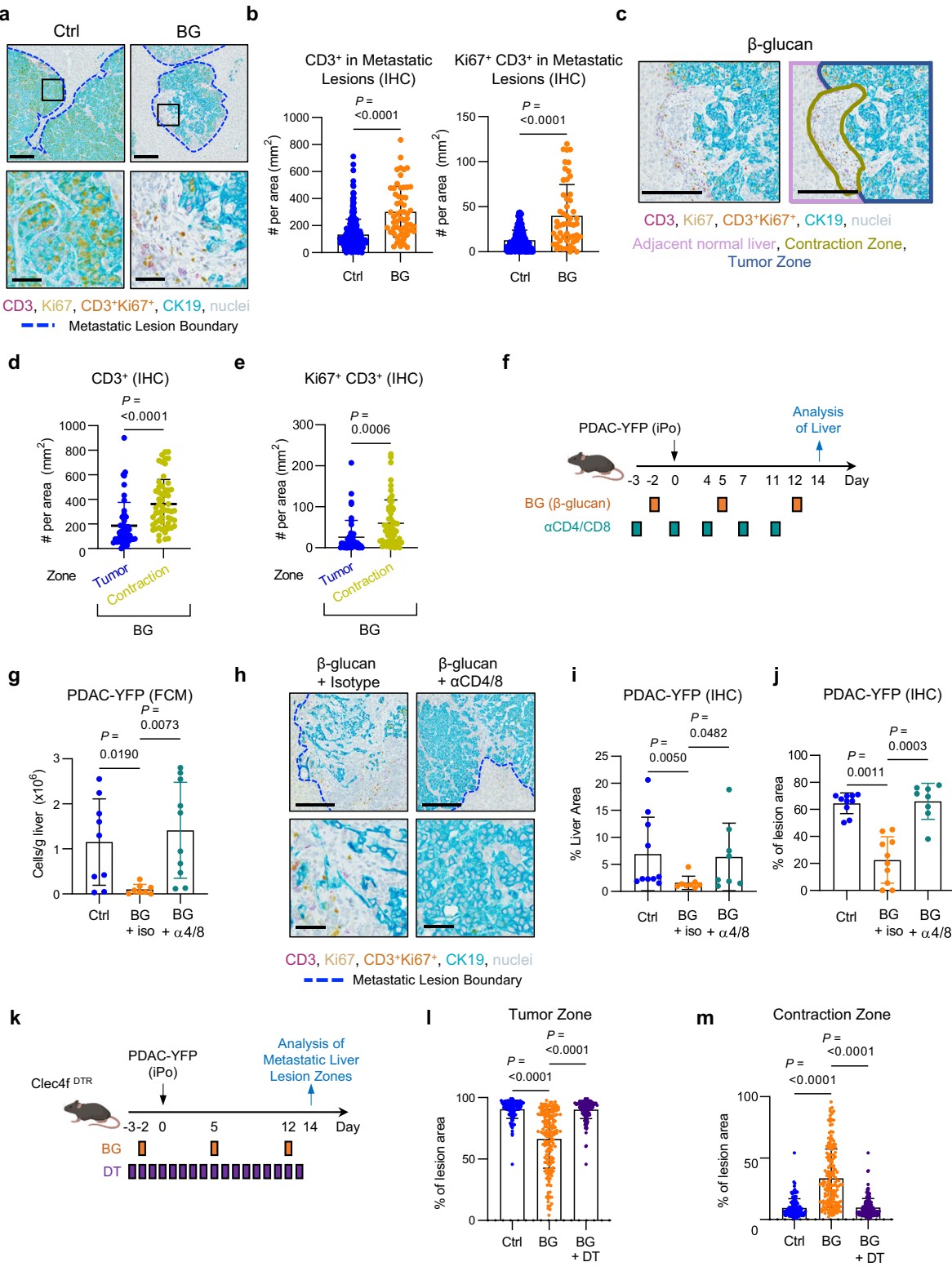

classical macrophage activation, characterized BG-treated *Spp1*+MΦs, we performed IHC for Nos2 on livers treated with BG with or without KC depletion using DT, macrophage depletion with CEL or T cell depletion with anti-mouse anti-CD4/CD8 antibodies. Indeed, BG triggered Nos2 expression in liver lesions, but this was lost upon depletion of KCs, BMDMs or T cells (Fig. 6k, l). In total, these data support a mechanism by which BG acutely activates KCs and T cells leading to re-education of BMDMs to control liver metastasis.

## BG treatment combines with anti-PD1 therapy to inhibit metastasis

Liver metastasis is associated with decreased benefit from immunotherapy with anti-PD1 (anti-PD1), an immune checkpoint blockade (ICB) that has shown activity across many solid malignancies[13–17]. Consistent with this clinical observation, liver metastasis is associated with peripheral immune tolerance mechanisms coordinated in part by liver macrophages[15]. Thus, we considered the possibility that the anti-metastatic and immunomodulatory activity of BG may sensitize liver

**Fig. 5 | β-glucan treatment stimulates T cell mediated anti-tumor immunity.** Study design for (**a**–**e**) is described in Fig. 1c. **a** Representative IHC images. Scale bars, 200μm (top) and 50μm (bottom). Metastatic lesions, blue dashed lines. **b** CD3$^+$ and CD3$^+$Ki67$^+$ cells detected by IHC and shown as cells per metastatic lesion area (mm$^2$). For CD3$^+$ cells in control (n = 289) and BG-treated (n = 57) tumor lesions were analyzed. For CD3$^+$Ki67$^+$ cells, control (n = 262) and BG-treated (n = 53) tumor lesions were analyzed. Unpaired two-tailed Welch's t test. **c** Representative IHC images. Contraction (olive green) and tumor (dark blue) zones are outlined. Scale bar, 200μm. Liver, light purple outline. **d**, **e** Quantification of CD3$^+$ (**d**) and CD3$^+$Ki67$^+$ (**e**) cells detected by IHC and shown as cells per zone area (mm$^2$) as drawn in (**c**). In (**d**) individual tumor zones (n = 57) and contraction zones (n = 47) and in (**e**) individual tumor zones (n = 57) and contraction zones (n = 46) were analyzed. Unpaired two-tailed Welch's t test. **f** Study design for (**g**–**j**). Data are representative of 3 experiments. **g** Quantification of PDAC-YFP tumor cells detected by FCM in control (n = 9), BG-treated (n = 8) and BG + anti-CD4/CD8 treated (n = 10) mice. One-way ANOVA with Sidak's test. **h** Representative IHC images. Scale bars, 200μm (top) and 50μm (bottom). Metastatic lesions, blue dashed lines. **i**, **j** PDAC-YFP tumor cells detected by IHC and shown as percent of liver area (**i**) in control (n = 10), BG-treated (n = 8) and BG + anti-CD4/CD8 treated (n = 8) mice and percent of metastatic lesion area (**j**) area in control (n = 10), BG-treated (n = 10) and BG + anti-CD4/CD8 treated (n = 8) mice. **k** Study design for **l**, **m**. **l**, **m**, Quantification of individual control (n = 138), BG-treated (n = 172) and BG + DT-treated (n = 175) tumor zones (**l**) and individual contraction zones (**m**) in livers from n = 10 mice per group. Kruskall-Wallis with Dunn's multiple comparisons (**i**, **j**, **l**, **m**). Mean ± SD is shown (**b**, **d**, **e**, **g**, **l**, **j**, **l**, **m**). BMDM, bone marrow-derived macrophage; iPo, intraportal; KC, Kupffer cell; DT, diphtheria toxin; Ctrl, control; BG, β-glucan; iso, isotype.

metastases to immunotherapy with anti-PD1. We first examined PD1 expression in liver metastatic lesions in mice. PD1 expression was uncommon in the normal liver but concentrated in metastatic lesions (Fig. 7a–d). However, BG (compared to control) was not associated with a significant increase in PD1 expression either in the liver (Fig. 7c) or metastatic lesions (Fig. 7d). We next tested the therapeutic activity of anti-PD1 in combination with BG when administered to mice with established liver metastases (Fig. 7e). Mice treated with combination therapy (BG plus anti-PD1) survived significantly longer compared to control mice (Fig. 7f). Notably, anti-PD1 monotherapy produced no improvement in survival whereas BG monotherapy was associated with a modest increase in survival (Fig. 7f). Together, these data show that BG sensitizes metastases to anti-PD1 immunotherapy.

Finally, we examined the biological effects of BG in combination with anti-PD1 in patients with advanced cancer. To do this, metastatic lesions were analyzed from patients enrolled on the IMPRIME-1 trial (NCT02981303). This Phase 2 study included patients with advanced triple-negative breast cancer (TNBC) or melanoma with metastatic disease (Supplementary Table S5). Eligible patients had previously progressed on front-line therapy and were treated with BG (odetiglucan, 4 mg/kg IV weekly) in combination with anti-PD1 (pembrolizumab, 200 mg IV q3weeks). Nine patients with metastases were biopsied before (pre-RX) and after (on-RX) 6 weeks of treatment. Biopsied tissues, including those from the liver, lymph node, neck, abdomen, and chest wall, were assessed histologically for tumor cells, macrophages, and T cells. On-treatment samples showed decreased tumor cell density (Fig. 7g, h). In addition, on-treatment samples contained a decrease in CD206$^+$ macrophages with a concomitant increase in CD80-expressing macrophages consistent with a shift in macrophage phenotype (Fig. 7i, j). Further, an increase in T cell infiltration, including CD4 and CD8 T cells, was seen in metastatic lesions in on-treatment compared to pre-treatment biopsies (Fig. 7k, l). Overall, these findings parallel our results seen in mice and support a mechanism in which BG redirects KCs to coordinate anti-tumor T cell-dependent immunity that then suppresses liver metastasis (Fig. 7m).

## Discussion

In this study, we investigated macrophages and their role in regulating liver metastasis. We show that liver resident macrophages, namely Kupffer cells, are a potential cellular target for triggering anti-metastatic activity. Metastatic lesions are frequently infiltrated by macrophages which are commonly associated with tumor progression. We found that both bone marrow-derived macrophages and Kupffer cells associate with liver metastases. However, only bone marrow-derived macrophages infiltrate metastatic lesions in the liver, whereas Kupffer cells remain restricted to the periphery of lesions. As a result, Kupffer cells infrequently contact cancer cells during metastasis. Despite this, our findings show that Kupffer cells can be activated and polarized by treatment with a β-glucan to inhibit metastasis. We found that β-glucans preferentially bound Kupffer cells in the liver.

However, anti-metastatic activity was dependent not only on Kupffer cells but also T cells. Consistent with this, Kupffer cells responded to treatment by upregulating genes involved in antigen presentation and associated with an interferon response. These findings are in line with prior reports using alternative approaches to polarize macrophages for the treatment of primary tumors and lung metastases[45]. However, strategies to leverage macrophages for the treatment of liver metastasis have remained ill-defined. Notably, liver metastasis is linked to immune resistance[15,46]. Our data now show that this resistance is reversible through macrophage polarization which sensitizes liver lesions to treatment with anti-PD1 immune checkpoint blockade. Taken together, our results identify a role for Kupffer cells in coordinating innate immune surveillance in cancer and underscore the prospect of myeloid agonists as a strategy to circumvent adaptive immune resistance in cancer immunotherapy.

β-glucan treatment stimulated Kupffer cells to coordinate anti-metastatic activity that was dependent on T cells. We found that treatment caused an immune reaction at the border of metastatic lesions which we termed a "contraction zone". Both Kupffer cells and T cells were required for the formation of this immune reaction which was notably devoid of tumor cells. In addition, T cells were found within the contraction zone, but Kupffer cells were excluded. In contrast, we detected T cell-Kupffer cell interactions in the liver that were increased by β-glucan treatment. Together, these findings raise the possibility that Kupffer cells may stimulate tumor-specific T cell responses through antigen presentation which occurs outside of metastatic lesions but within the liver. Consistent with this, we found that β-glucan treatment increased Kupffer cell expression of *Irf7*, which is known for its critical role in interferon responses that govern the induction of T cell immunity[47]. β-glucan treatment also increased the expression of *Isg15* in Kupffer cells. *Isg15* is an interferon-induced ubiquitin-like protein with immunomodulatory potential capable of enhancing APC function[48]. Thus, these findings support a role for polarizing Kupffer cells to invoke anti-metastatic responses and emphasize the importance of bridging innate and adaptive immunity for cancer immunotherapy.

The liver is a lymphoid organ that is central to establishing peripheral immune tolerance[49]. In cancer, the liver is continuously exposed to malignant cells as well as soluble factors and antigens released by tumors and commensal organisms (e.g. bacteria) originating from the gut[50]. These events trigger alterations in the liver that impact its biological state[51,52]. Resident liver macrophages, namely Kupffer cells, are well-recognized for their role in establishing peripheral immune tolerance[53–56]. Kupffer cells maintain tolerance through the production of IL-10 and prostaglandins, by acting as incompetent APCs, and through the expansion of Tregs[53–55]. Kupffer cells have also been suggested to mediate tolerance in cancer via Fas-dependent apoptosis of tumor-specific CD8$^+$ T cells[15]. However, it is known that macrophages may also be induced with immunostimulatory properties[57,58]. Despite this, recent literature suggests that

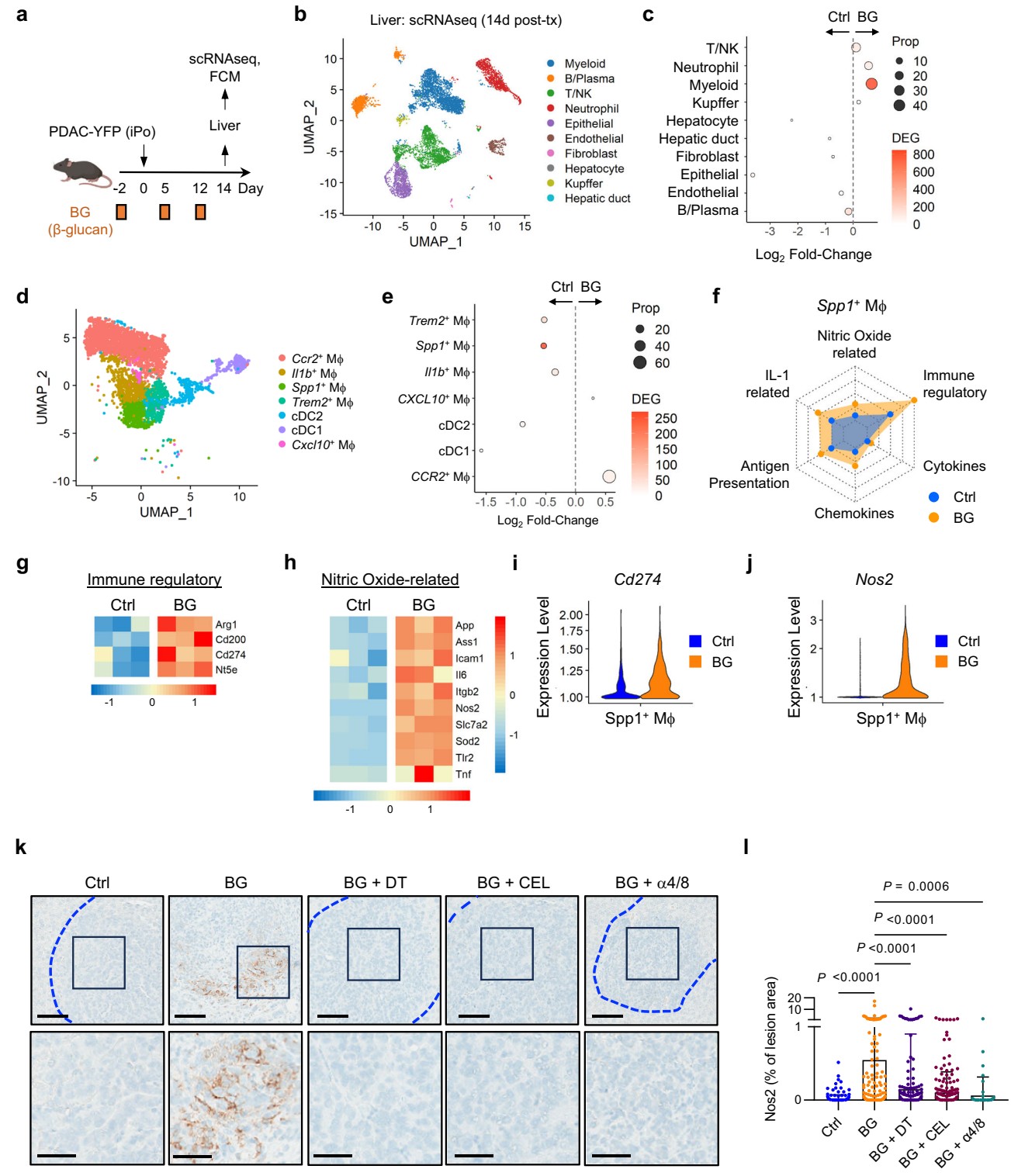

Nos2, nuclei ----- Metastatic Lesion Boundary

macrophage pliability is more permissive for bone marrow-derived macrophages than resident macrophages, such as Kupffer cells. This hypothesis is founded on the principal that plasticity undermines the requirement of resident macrophages to maintain tissue homeostasis[59,60]. Consistent with this, resident macrophages produce an attenuated inflammatory response to external stimuli. We also found that Kupffer cells upregulated fewer inflammatory genes in response to β-glucan treatment than bone marrow-derived macrophages. However, the requirement of Kupffer cells for β-glucan

treatment efficacy suggests that they have sufficient pliability to be instructed with anti-metastatic activity.

In our experiments, we found that β-glucan treatment delivered before or after metastatic challenge reduced metastasis raising the possibility that Kupffer cells may be imprinted with anti-metastatic activity. This is consistent with prior literature showing the capacity of β-glucans to rewire macrophages with trained immunity[28,61], a process involving epigenetic and metabolic reprogramming that results in altered responses to secondary challenge[62]. We found that β-glucan

**Fig. 6 | Kupffer cells and T cells coordinate BMDM activation. a** Study design for (**b**–**j**). **b** UMAP of liver populations identified by scRNAseq in control (n = 3) and BG (n = 3)-treated livers. **c** Dot plot of Log$_2$ Fold-Change in liver population frequencies in BG-treated livers compared to control livers. Size represents proportion of total cells and color indicates number of differentially expressed genes in BG versus control. **d** UMAP of BMDM/cDC populations identified by scRNAseq. **e** Dotplot of Log$_2$ fold change in BMDM population frequencies in BG-treated livers compared to control livers. Size represents proportion of total cells and color indicates number of differentially expressed genes in BG versus control. **f** Radar plot showing selected gene sets expressed by *Spp1*⁺MΦs in control (blue) and BG (orange) treated livers. Axis represents the mean TPM of selected gene set-associated genes with range 0 to 172 TPM. **g, h** Heatmaps of immune regulatory (**g**) and nitric oxide-related (**h**) gene set associated genes, upregulated in *Spp1*⁺MΦs from control and BG-treated livers. Scale bars indicate log$_2$[TPM] scaled by row across samples. **i, j** Violin plots of *Cd274* (**i**) and *Nos2* (**j**) expression by *Spp1*⁺MΦs in control and BG-treated livers. **k** Representative images of livers from control mice or mice treated with BG, BG + DT, BG + CEL, or BG + a4/8 stained using IHC to detect NOS2 (brown) and nuclei (blue, hematoxylin). The bottom panel highlights the lesion interior. Scale bars, 100 μm (top) and 50 μm (bottom). Metastatic lesions, blue dashed lines. **l** Quantification of NOS2⁺ staining detected by IHC and shown as percentage of individual control (n = 248), BG-treated (n = 173), BG + DT (n = 350), BG + CEL (n = 260) and BG + α4/8 (n = 61) liver lesion area. Data represents the combination of three independent experiments with control (n = 5), BG-treated (n = 6), BG + DT (n = 8), BG + CEL (n = 5), and BG + α4/8 (n = 5) biological replicates. One-way ANOVA with Sidak's multiple comparisons test. Mean ± SD is shown. iPo, intraportal; KC, Kupffer cell; BG, β-glucan; scRNAseq, single cell RNA sequencing; MΦ, macrophage; cDC, conventional dendritic cell; Ctrl, control; DT, diphtheria toxin; CEL, clodronate encapsulated liposomes; a4/8, anti-mouse anti-CD4/anti-CD4 monoclonal antibody.

treatment rapidly altered the phenotype of Kupffer cells by down-regulating expression of CD206, a molecule commonly associated with immunosuppressive myeloid cells[63]. In addition, Kupffer cells upregulated CD38 which is associated with myeloid cells of an immunostimulatory phenotype[61,64]. Notably, this shift in Kupffer cell biology was maintained at 2 weeks after β-glucan treatment. As Kupffer cells are long-lived resident liver macrophages, modulating their biology may produce a more durable shift in the immunological state of the liver compared to transient activation of liver-infiltrating macrophages that is seen with myeloid agonists (e.g. CD40) targeting bone marrow-derived macrophages[65–67]. Thus, therapeutically intervening on Kupffer cells with β-glucan treatment may offer a unique opportunity to reprogram the liver from immune-suppressive to immune-permissive.

Liver metastasis correlates with decreased efficacy of immune checkpoint blockade[13–17,68]. Disseminated cancer cells that invade the liver may provoke immune tolerance mechanisms that hamper anti-tumor immunity and in doing so, limit the efficacy of cancer immunotherapy. Consistent with this, liver macrophages have been implicated as determinants of immune tolerance invoked by liver metastasis due to their capacity to trigger apoptosis of tumor-specific CD8⁺ T cells[15]. Our findings are consistent with these observations as we show that anti-PD1 is ineffective against liver metastasis in models of pancreatic cancer. Prior studies have investigated strategies to deplete myeloid subsets as an approach to remove mechanisms of immunological tolerance triggered by liver metastasis[18,69,70]. However, myeloid depletion strategies are susceptible to the emergence of compensatory myeloid responses that paradoxically support tumor outgrowth[71]. In contrast, our findings in mouse models show that macrophages can be leveraged for their anti-metastatic potential and capacity to condition tumors with sensitivity to anti-PD1 immune checkpoint blockade. Further, in patients with triple-negative breast cancer and advanced melanoma treated with a β-glucan (odetiglucan) in combination with anti-PD1 (pembrolizumab) therapy (IMPRIME-1 Study), we observed a shift in the polarization of macrophages with a corresponding increase in infiltrating T cells in metastatic lesions[72,73]. We also found that the effects of β-glucan were not liver specific with anti-metastatic activity also observed in a model of lung metastasis. Together, these data suggest polarization of resident tissue macrophages using β-glucans as a potential strategy to sensitize metastases to immunotherapy.

Beta-glucan therapy is under clinical investigation as a strategy to enhance the efficacy of immune checkpoint blockade with anti-PD1 therapy (NCT05159778) Our study identifies a role for Kupffer cells in coordinating cancer immunosurveillance and shaping outcomes to immunotherapy. Further, these findings support the development of interventions targeting tissue-resident macrophages as a strategy to reverse resistance in cancer immunotherapy and improve outcomes for patients with metastatic disease.

## Methods

### Study design
All animal studies were conducted in accordance with guidelines from the Animal Care and Use Committee of the University of Pennsylvania. All studies of human biospecimen were conducted in accordance with recognized ethical guidelines including the Belmont Report, CIOMS, Declaration of Helsinki, and U.S. Common Rule.

### Cell lines
PDA.8572 (PDAC−YFP) and PDA.69 (PDAC.69) cell lines were used for intrasplenic, intraportal, and tail-vein injections. PDA.8572 cell line was derived from a PDAC tumor that arose spontaneously in a female KPCY mouse, as previously described[51,58,74]. PDA.69 was derived from a PDAC tumor that arose spontaneously in a female KPC mouse, as previously described[51]. Cell lines were maintained in DMEM (Corning) supplemented with 10% fetal bovine serum (FBS, VWR), 83 μg/ml gentamicin (Thermo Fisher), and 1% GlutaMAX (Thermo Fisher), henceforth referred to as DMEM complete media. Cells were cultured at 37 °C, 5% CO$_2$. Cell lines that had been passaged fewer than 16 times were used for experiments, and trypan blue staining was used to ensure that cells with >90% viability were used for studies. Cell lines were tested routinely for *Mycoplasma* contamination at the Cell Center Services Facility at the University of Pennsylvania and all lines used tested negative for contamination.

### Mice
C57BL/6 J (#000664), Clec4f-cre (C57BL/6J-*Clec4f*$^{tm1(cre)Glass}$/J, #033296) and Rosa-DTR (C57BL/6-*Gt(ROSA)26Sor*$^{tm1(HBEGF)Awai}$/J, #007900) mice were procured from the Jackson Laboratory. Clec4f-cre⁺/⁺ mice were crossed with Rosa-DTR$^{flox/flox}$ to generate Clec4f$^{DTR}$ (Clec4f-cre⁺/⁻ Rosa-DTR$^{flox/+}$) mice for use in experiments. All mice were bred and maintained under pathogen-free conditions in a barrier animal facility at the University of Pennsylvania. Animal protocols were reviewed and approved by the Institute of Animal Care and Use Committee of the University of Pennsylvania (protocol number 803605). Euthanasia was performed using CO$_2$ inhalation following AVMA and institutional guidelines. Mice were monitored three times per week and euthanized based on defined criteria including loss of ≥ 20% body weight, body condition score ≥ 2, lethargy, or other signs of distress. Metastatic liver lesions were not grossly measurable and thus tumor measurements were not routinely taken.

### Animal studies
For all studies, mice of similar age and sex were used. Both male and female mice between 8 to 12 weeks of age were used, except for overall survival and scRNAseq studies where only female mice were used. Mice were co-housed and enrolled in a randomized, unblinded fashion. Sample sizes were calculated based on the number of mice needed for statistical analysis determined from pilot studies.

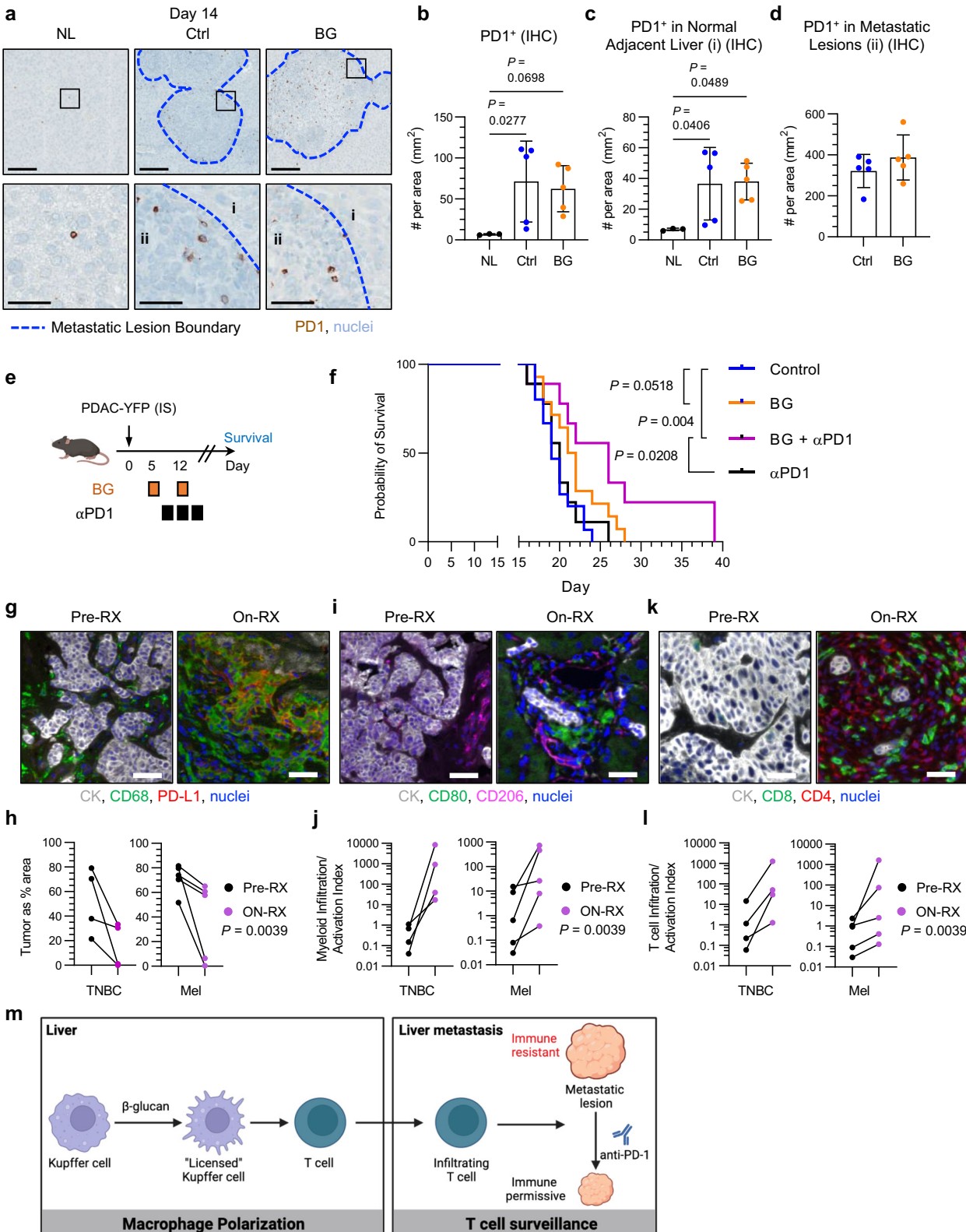

For intraportal (iPo) injection of tumor cells, mice were anesthetized with continuous isoflurane. Analgesics (e.g. buprenorphine) were administer for pain control. Briefly, the abdomen was shaved and sterilized, and depth of anesthesia was assessed prior to performing a median laparotomy. The incision (10-15 mm) was used to expose the peritoneal cavity. A similarly sized incision was then made in the peritoneum and the incision was held open with an Agricola retractor

(Roboz). The intestines were exteriorized to expose the portal vein. Intestinal hydration was maintained with pre-warmed (37 degree) sterile PBS throughout the procedure. Pancreatic tumor cells ($2 \times 10^5$ 8572YFP or $5 \times 10^5$ PDAC.69 suspended in 100 uL sterile PBS) were injected into the portal vein via a 30-gauge needle. Successful injection was confirmed by blanching of the liver with minimal leakage. Hemorrhage was controlled with pressure applied with sterile gauze

**Fig. 7 | BG treatment and anti-PD1 therapy combine to inhibit liver metastasis.**
**a** Representative images of livers from tumor-free mice (Normal Liver, NL, left) and mice treated with control (middle) and BG (right) at 14 days after iPo injection of PDAC-YFP cells. Tissues are stained using IHC to detect PD1 (brown) and nuclei (blue, hematoxylin). Scale bar, 200 µm (top) and 50 µm (bottom). **b–d** PD1[+] cells detected by IHC in liver on Day 14 are shown as (**b**) cells per liver area, (**c**) cells per liver area in normal liver (n = 3) and after iPo injection of PDAC-YFP without treatment (n = 5) or after treatment with BG (n = 5), and (**d**) cells per metastatic lesion area (mm²). Kruskal-Wallis test with Dunn's multiple comparisons testing (**b, c**) and unpaired two-tailed Welch's t test (**d**). **e** Study design for (**f**). Data are representative of 2 independent experiments. **f**, Kaplan-Meier plot of mice following IS injection of PDAC-YFP cells in control (n = 15), BG-treated (n = 14), BG + anti-PD1 (n = 9) and anti-PD1 (n = 9) treated mice. Log-rank test. **g–l** Biopsies of metastatic lesions from patients with TNBC (n = 4) and melanoma (Mel, n = 5) were collected prior to (Pre-RX, left) and after 6 weeks of treatment (On-RX, right) with β-

glucan (Odetiglucan) + anti-PD1 (pembrolizumab). **g** Representative IF images of paired liver biopsies stained for cytokeratin (CK, white), CD68 (green), PD-L1 (red), and nuclei (blue). Scale bars, 20 µm. **h** Quantification of tumor area pre- and on-treatment for individual patients. Scale bars, 20µm. **i** Representative IF images of paired liver biopsies stained for cytokeratin (white), CD80 (green), CD206 (pink), and nuclei (blue). Scale bars, 20 µm. **j** Quantification of myeloid infiltration/activation index for individual patients. **k** Representative IF images of paired liver biopsies stained for cytokeratin (white), CD8 (green), CD4 (red), and nuclei (blue). **l** Quantification of T cell infiltration/activation index for individual patients. **m** Conceptual model. β-glucan treatment activates Kupffer cells, which in turn recruit T cells to metastatic lesions resulting in decreased tumor cell proliferation and sensitization of metastatic lesions to anti-PD1 immunotherapy. Mean ± SD is shown (**b, c, d**). IHC, immunohistochemistry; IS, intrasplenic; IF, immuno-fluorescence; TNBC, triple-negative breast cancer.

and hemostatic flour/gauze (Avitene). The intestines were then returned to the abdomen; the peritoneum was closed with suture; and the skin was closed with wound clips. Buprenorphine was administered for 48 h following the surgery. For overall survival studies, mice were monitored twice weekly and euthanized for development of ascites or decreased activity. For endpoint studies, mice were euthanized at 2, 12, or 14 days after surgery, as indicated.

For intrasplenic (IS) injections of pancreatic tumor cells, mice were similarly prepared for surgery as for intraportal injection surgeries. Briefly, an incision (5–10 mm) in the upper left quadrant of the abdomen exposed the peritoneal cavity. A similar incision was then made in the peritoneum. The spleen was exteriorized and injected with pancreatic tumor cells (5 × 10⁵) suspended in 100 uL sterile PBS through a 30-gauge needle. Successful injection was confirmed by whitening of the spleen and minimal leakage. Splenectomy along with a pancreatectomy involving resection of the pancreatic tail was then performed using cauterization and hemostatic flour and gauze (Avitene) to control hemorrhage. All organs were then returned to the peritoneal cavity and the peritoneum was closed with 5–0 PDS II violet suture (Ethicon). The skin was closed with wound clips (Braintree Scientific). After the procedure, mice were similarly followed as described for intraportal injection surgeries.

For intravenous (IV) injection of pancreatic tumor cells, 5 × 10⁵ PDAC-YFP cells were suspended in sterile PBS and injected into the tail vein of mice. Prior to injection, mice were warmed in their cage on a slide warmer for 30 minutes. Immediately prior to injection, each tail was heated under a heat lamp, sterilized, and cells were injected via a 30-gauge needle.

Antibodies administered to mice included anti-CD4 (GK1.5, 0.2 mg), anti-CD8 (2.43, 0.2 mg), anti-PD1 (RMP1-14, 0.2 mg), and rat isotype control (LTF-2, 0.2 mg) antibodies and were suspended in 200 µl sterile PBS for administration. The abdomen of mice was sterilized, and antibodies were injected into the peritoneum via a 30-gauge needle. All in vivo antibodies were sourced from BioXCell. To deplete liver macrophages, clodronate-encapsulated liposomes (CEL, Liposoma, 200uL) were administered by intraperitoneal (IP) injection according to the manufacturer's protocol. To deplete Kupffer cells, diphtheria toxin (DT, 200 ng) was delivered IP to Clec4f^DTR mice. β-glucan (BG, odetiglucan, Hibercell, 1.2 mg) is a clinical grade soluble, β−1,3/1,6 glucan derived from *Saccharomyces cerevisiae*[75]. BG was suspended in 100µL PBS and administered IV to mice weekly, unless otherwise noted. For overall survival studies, mice received up to seven doses of BG. Detailed information on antibodies and reagents used in experiments can be in found in Supplementary Table S1.

## Immunohistochemistry

Dissected tissues were fixed in 10% formalin for 48 hours at room temperature, then washed twice with PBS and stored in 70% ethanol at 4 °C until embedded in paraffin. Five-micron formalin-fixed paraffin-

embedded (FFPE) tissues sections were used for staining. Hematoxylin and eosin (H&E) stains were done manually. Automated immunohistochemistry staining was performed on FFPE sections using a Ventana Discovery Ultra automated slide staining system (Roche). The reagents used are detailed in Supplementary Table S2. Images were acquired using an Aperio CS2 scanner system (Leica). For quantification of cell populations, Visiopharm software (version 2020.01.1.7332) was used to analyze the whole tissue of interest. Cells were detected and classified based on colorimetric characteristics within regions of interest (ROI) using custom algorithms.

## Human samples

Biopsy samples from human PDAC liver metastasis lesions were collected from patients enrolled in the Pancreatic Cancer Action Network (PanCAN) Know Your Tumor (KYT) program[76] and were provided by PanCAN. For this program, patients self-refer to the PanCAN call center and enroll through an institutional review board (IRB)-approved protocol. The study was approved by the New England IRB. Prior to enrollment, written informed consent was obtained from each participant or participant's guardian. Research was conducted in accordance with recognized ethical guidelines including the Belmont Report, CIOMS, Declaration of Helsinki, and U.S. Common Rule. Tissues were collected as surgical specimens or core biopsies using an 18-20 gauge needle.

Paired biopsies from non-targeted lesions from liver (n = 2), lymph node (n = 4), neck (n = 1), chest wall (n = 1), and abdomen (n = 1) were collected from patients enrolled on the IMPRIME-1 trial (NCT02981303), a multicenter, open-label, Phase 2 study of soluble β-glucan (odetiglucan) and pembrolizumab (KEYTRUDA®, pembrolizumab). Written informed consent was obtained from each participant or participant's guardian. The IMPRIME-1 trial enrolled patients across multiple sites and the study was approved by a central Western Institutional Review Board and local institutional review boards including the UCLA Office of Human Research Protection Program and the Mayo Clinic Institutional Review Board (Supplementary Table S5). Pembrolizumab is a humanized mAb against PD1. Patients enrolled in IMPRIME-1 included (i) unresectable Stage III or metastatic (Stage IV) melanoma not amenable to local therapy, and irrespective of PD-L1 status, with objective radiographic or clinical disease progression after PD-1/PD-L1 +/- anti-CTLA-4 inhibitor therapy and (ii) metastatic (Stage IV) triple-negative breast cancer (TNBC) irrespective of PD-L1 status and after at least one 1 line of chemotherapy for metastatic disease. Inclusion criteria were age ≥ 18 years old; resolution of all previous treatment-related toxicities to Grade 1 severity or lower, except for stable sensory neuropathy (less than or equal to Grade 2) or alopecia; recovery from any complication and/or toxicity from major surgery or radiation therapy of >30 Gy; measurable metastatic disease based on RECIST v1.1; peripheral blood levels of IgG anti-β-glucan antibody (ABA) of > 20 mcg/

mL; willing to consider providing fresh tissue for biomarker analysis; ECOG performance status 0 or 1; life expectancy ≥ 3 months; and adequate end organ function. Patients were excluded for disease suitable for local therapy with curative intent; prior investigative therapy within 4 weeks of initiating treatment; immunodeficiency or receiving systemic steroid therapy and other form of immunosuppressive therapy within 7 days of initiating treatment; active tuberculosis, Hepatitis B or C; history of HIV; history of clinically severe autoimmune disease or organ transplant; history of ocular melanoma; hypersensitivity to baker's yeast; prior exposure to Betafectin or odetiglucan; hypersensitivity to pembrolizumab or any of its excipients; prior anti-cancer monoclonal antibody (except immune checkpoint inhibitor for melanoma subjects) within 30 days prior to start of study treatment; prior chemotherapy, targeted therapy or radiation therapy within 2 weeks of initiating treatment; receiving transfusion of blood products within 4 weeks of initiating treatment; secondary active malignancy except basal cell carcinoma of skin, squamous cell carcinoma of skin, or in situ cervical cancer; active central nervous system metastases and/or carcinomatous meningitis; active autoimmune disease requirement treatment in past 2 years; pneumonitis; history of interstitial lung disease; active infection or receiving live-virus vaccination within 30 days of initiating treatment; and clinically significant cardiovascular disease. Patients with TNBC were excluded if received prior therapy with an anti-PD-1, anti-PD-L1, anti-CTLA-4, or anti-PD-L2 agent. Patients received odetiglucan (4 mg/kg IV days 1, 8, 15 of each 3-week cycle) plus pembrolizumab 200 mg on day 1 of each cycle. Biopsies were collected during screening and prior to cycle 3 (6 weeks post-treatment initiation) to assess immune activation in tumor tissue.

## Immunofluorescence

Biopsy samples obtained from patients were formalin fixed and paraffin embedded at clinical sites per standard protocols. Slides were deparaffinized, rehydrated and heat mediated antigen retrieval was conducted with AR6 buffer (Akoya). Slides were blocked with Roche diagnostics antibody diluent (Fisher) and stained with primary antibodies in a humidified chamber on a gyrating rocker for 1 hour at 110 rpm. HRP-conjugated secondary antibodies were added for 10 minutes and then incubated an additional 10 minutes with OPAL detection dye (Akoya). For multiplex staining, this process was repeated starting at the antigen retrieval stage. Antibodies used are listed in Supplementary Table S3. Multispectral Images were captured on the Vectra 3 imaging system (Akoya). Spectral unmixing and image analyses were performed using Inform Tissue Finder software (Akoya). Tumor regions of interest (ROIs) were defined, and cell segmentation was performed on the images. Imaging data containing marker intensities for each segmented cell was then converted into .csv files and imported into Flowjo for immune phenotyping. For each marker, the number of cells per sample was determined and used to calculate (i) tumor content per ROI area, (ii) macrophage infiltration/activation index, and (iii) T cell infiltration/activation index. Macrophage activation/infiltration index was calculated by the equation [(total macrophages/total tumor cells)*((CD80+ macrophages/total macrophages)/(CD206+ macrophages/total macrophages))]. T cell activation/infiltration index was calculated by the equation [(total T cells/total tumor cells)*((Ki67+ T cells/total T cells)+(Granzyme B+ T cells/total T cells))].

## Flow cytometry

For experiments assessing tumor burden, mice were euthanized and then the portal vein and inferior vena cava were severed to drain the blood from the liver and lungs. For experiments assessing resident immune cell phenotype, mice were euthanized, and then the lungs and liver were perfused with PBS through the portal vein, inferior vena cava, and the heart. Successful perfusion was achieved by complete blanching of the organ. Liver and lungs were removed and rinsed in

DMEM complete media and then minced with micro-dissecting scissors into small pieces in DMEM containing collagenase (1 mg/ml, Sigma-Aldrich) and DNase (150 U/ml, Roche). Tissues were then incubated at 37 °C for 30 min with intermittent agitation, filtered through a 70μm nylon strainer (Corning), and washed three times with FACS buffer (PBS with 2% FBS and 0.2 mM EDTA). Lysis of red blood cells was performed using ACK lysing buffer (Quality Biological) at room temperature for 5 minutes. Cells were then washed with FACS buffer. Cell suspension was then passed through a 40μm strainer. Peripheral blood was collected from the tail vein of mice into a capillary tube. Red blood cells were lysed in ACK lysis buffer. Cells were then counted using a BioRad TC20 automated cell counter and stained using Aqua dead cell stain kit (Life Technologies) according to the manufacturer's protocol. For immune cell subset characterization, cells were washed twice with FACS buffer and stained with appropriate antibodies (Supplementary Table S4). Tumor burden was quantified by flow cytometry (FCM) by numerating PDAC-YFP cells based on endogenous YFP expression. Samples were examined using a FACS Canto II (BD Biosciences) and analysis was performed using Flowjo (FlowJo, LLC, version 10.8).

## Single cell RNA sequencing

Tissues were processed in RPMI supplemented with 10% fetal bovine serum, 83 μg/ml gentamicin, and 1% GlutaMAX, henceforth referred to as RPMI complete media, digested in RPMI containing collagenase and DNase, and washed with RPMI complete media. After passage through a 40μm strainer, a single-cell suspension was achieved. Dead cells were removed using the Dead Cell Removal Kit (STEMCELL Technologies). CD45+ cells were then isolated with the CD45+ isolation kit (STEMCELL Technologies). To increase the recovery of KCs, the flow through from the CD45+ isolation kit was then layered over Ficoll Plaque Plus (Millipore Sigma) and spun at 2200 rpm for 20 minutes. Cells from the buffy coat were then combined with the CD45+ cells and passed through a 35μm strainer to ensure a single-cell suspension.

Next-generation sequencing libraries were prepared using the 10x Genomics Chromium Single Cell 3' Reagent kit v3.1 per manufacturer's instructions. Libraries were uniquely indexed using the Chromium dual Index Kit, pooled, and sequenced on an Illumina NovaSeq 6000 sequencer in a paired-end, dual indexing run. Sequencing for each library targeted 20,000 mean reads per cell. Data were then processed using the Cellranger pipeline (10x genomics) for demultiplexing and alignment of sequencing reads to the mm10 transcriptome with the included introns parameter and creation of feature-barcode matrices.

## Single cell sequencing analysis

scRNAseq data analysis was performed using Seurat (version 4.1.1; https://satijalab.org/seurat/) and R (version 4.1.2)[77–80]. Data were filtered to include cells with at least 200 genes and genes present in at least 3% of cells were included in the analysis. Cells were excluded if they contained reads for greater than 2500 genes or if more than 30% of the reads aligned to mitochondrial genes. The NormalizeData function with the LogNormalize method and a scale factor of $10^4$ was used to normalize the data. The 'vst' selection method was used to identify highly variable genes (n = 2000) and the ScaleData function was used to scale and center genes to a mean of 0. The RunPCA function was applied using previously identified genes for linear dimensionality reduction. Batch correction was performed using integration. Cell clusters were identified using the FindNeighbors and FindClusters functions with a resolution of 0.5. The RunUmap feature was used to perform non-linear dimensionality reduction to visualize clusters in a 2-dimensional space. Cluster markers were identified using the FindAllMarkers function, testing for differentially expressed genes between cells in a single cluster as compared to all other clusters. Clusters were defined by a manual review of cluster-specific differentially expressed genes and the SingleR tool. Differentially

expressed gene pseudobulk analysis was performed using DESeq2[81]. Gene set enrichment analysis was performed using GSEA (http://software.broadinstitute.org/gsea/index.jsp). Radar plots were generated using radar-chart in R.

**Statistical analysis**

Sample sizes were estimated based on pilot experiments conducted in the laboratory and were selected to provide sufficient numbers of mice in each group to yield a two-sided statistical test, with the potential to reject the null hypothesis with a power of (1-β) of 80%, subject to α = 0.05. Prism (GraphPad Software, version 9.4) was used to determine statistical significance. Welch's T-test, one-way ANOVA with Tukey's test or Sidak's test, two-way ANOVA with Tukey's test or Sidak's test, Logrank test, and Kruskal-Wallis test with Dunn's multiple correction were used for analysis. Survival analysis was assessed using Kaplan–Meier plots with significance determined using Logrank test. A P value < 0.05 was considered significant. Experiments were randomized during enrollment and unblinded during study conduct and data analysis. Statistical analysis was performed using Prism (GraphPad Software, version 9.2.0).

**Reporting summary**

Further information on research design is available in the Nature Portfolio Reporting Summary linked to this article.

## Data availability

The raw data generated in this study are provided in the Source Data file. The raw and processed scRNAseq files have been deposited in Gene Expression Omnibus (GEO) under accession numbers GSE235318. Source data are provided with this paper.

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

## Acknowledgements

The authors would like to thank the Center for Applied Genomics (CAG) at The Children's Hospital of Philadelphia (CHOP) and the Molecular Pathology and Imaging Core (University of Pennsylvania) for technical support; I. E. Brodsky, M. Haldar, H. Bassiri, and L.C. Eisenlohr for discussions and advice; and the Pancreatic Cancer Action Network for provision of liver tissue sections collected from patients with pancreatic cancer. Biorender.com was used to generate model figures. This research was supported by the National Institutes of Health grants F30-CA257287-01 (S.K.T.), T32 HL007439 (S.K.T.), R01-CA197916 (G.L.B.), R01-CA245323 (G.L.B.), and U01 CA224193 (G.L.B.), the Cancer Research Institute Irvington Fellowship (M.L.S), and the 2017 Pancreatic Cancer Action Network Precision Medicine Targeted Grant 17-85-BEAT (G.L.B.). M.M.W. is a Damon Runyon Physician-Scientist supported (in part) by the Damon Runyon Cancer Research Foundation (PST-34-21). Additional funding support was provided by HiberCell, Inc. Odetiglucan (Imprime PGG) is owned by Hibercell, Inc., which manages its distribution.

## Author contributions

S.K.T.: Conceptualization, formal analysis, investigation, methodology, writing–original draft, writing–review and editing. S.C-B., B.H., M.U., D.P., K.M., & D.D.: Formal analysis, investigation. H.C., C.R.C., & M.L.S.: Investigation, M.C., J.D. & N.B.: Conceptualization. M.M.W.: Writing review and editing, formal analysis, investigation. G.L.B.: Conceptualization, supervision, writing–original draft, writing–review and editing. All authors critically revised the manuscript.

## Competing interests

G.L.B. reports prior or active roles as a consultant/advisory board member for Boehinger Ingelheim, Adicet Bio, Aduro Biotech, AstraZeneca, BiolineRx, BioMarin Pharmaceuticals, Boehinger Ingelheim, Bristol-Myers Squibb, Cantargia, Cour Pharmaceuticals, Genmab, HiberCell, HotSpot Therapeutics, Incyte, Janssen, Legend Biotech, Merck, Monopteros, Molecular Partners, Nano Ghosts, Opsona, Pancreatic Cancer Action Network, Seagen, Shattuck Labs, and Verastem, and; reports receiving commercial research grants from Alligator Biosciences, Arcus, Bristol-Myers Squibb, Genmab, Gilead, Halozyme, HiberCell, Incyte, Janssen, Newlink, Novartis, and Verastem,. G.L.B. is an inventor of intellectual property (U.S. patent numbers 10,640,569 and 10,577,417) and recipient of royalties related to CAR T cells that is licensed by the University of Pennsylvania to Novartis and Tmunity Therapeutics. N.B., J.D., B.H. are employees of Hibercell, Inc. M.U. is a previous employee of Hibercell and current employee of OncXerna. N.B., J.D., B.H., M.U. are Hibercell stockholders. N.B., M.U. are an inventors of intellectual property that is licensed to Hibercell. M.U. holds OncXerna and Rigel stock. M.C. is an employee of Merck & Co., Inc and holds Merck stock. M.M.W. reports prior or active roles as a consultant for Nanology. The remaining authors declare no competing interests.
