## [Peer Review File · Nature Communications]

REVIEWER COMMENTS

Reviewer #1 (expert in PDAC metastasis):

Thomas et al. described their study showing that β -glucan triggered activation of liver Kupffer cells (KC), suppressed cancer cell proliferation, and invoked anti-metastatic T cell immunity in pancreatic cancer mouse liver metastasis models. They also presented their analysis of pre- and on-treatment biopsy tumor specimens from the clinical trial that tested BG (odetiglucon) in combination with anti-PD-1 antibody in a Phase 2 study included patients with advanced triple negative breast cancer (TNBC) or melanoma with metastatic disease. It is nice to see a study combining both mouse model and human specimen analyses. Using a mouse model with a genetic depletion of KC is also a strength of this manuscript. However, the role of Kupffer cells in mediating treatment response to BG would need more data and a better mouse model to establish.

In the beginning of the manuscript, authors showed that CEL depletion of macrophages produced a significant decrease in metastatic tumor burden in the liver. However, in the remaining study, the manuscript is focused on showing depletion of macrophages leads to the abolish of antitumor effect of BG. Although these two opposite effects of macrophage depletion are not conflicted as the manuscript hypothesizes that macrophages can be reprogrammed/repolarized into antitumor macrophages, the data showing the protumor activity of CEL depletion of macrophages may not add much to the main theme of the macrophages and should be considered to move to the supplemental data.

Depletion of either BMDM or KC would decrease the treatment effects on the liver metastatic burdens. BG is also effective in the lung metastasis model. Therefore, the mouse experiment results doesn't seem to be able to establish the role of KC in mediating the treatment effect of BG.

Liver metastasis in the splenic vessel injection mouse model is not formed spontaneously. Authors should consider using the pancreatic orthotopic mouse model and examining the spontaneous liver metastases to study the treatment effect of BG with or without KC depletion.

Figure 4e-I showed that macrophage depletion produced a significantly higher metastatic burden in the liver of BG treated mice. In this experiment, a control group should be added to compare liver metastasis burden in BG treated mice vs. liver metastasis burden in untreated, CEL depleted mice.

For pre- and on-treatment biopsy specimen analysis, the results on liver metastases should be compared with other metastatic lesions. How the tumor areas are quantified is not described. Microscopically measuring the tumor areas is not previously validated. Radiographic responses should be used to characterize the antitumor effect of BG.

Clinical trial information should be provided in more detail, including patient demographic and clinicopathologic information. Radiographic responses should also be described.

Reviewer #2 (expert in PDAC therapy):

In the manuscript entitled "Kupffer cells coordinate anti-metastatic activity and immune resistance in the liver", Stacy Thomas and colleagues study the importance of macrophage polarity in metastasis and identify Kupffer cells as a novel therapeutic target to reverse cancer immune resistance.

Main comment

The rationale for the study seems reasonable as it is currently not clear how macrophage polarization could be an approach to intervention in metastasis. They report that macrophages

respond to β -glucan treatment by inhibiting liver metastases. β -Glucan induced activation of macrophages in the liver (Kupffer cells), suppressed cancer cell proliferation, and elicited productive anti-metastatic T-cell immunity in mouse models of pancreatic cancer. Although these results are quite interesting, the manuscript has numerous deficiencies that preclude an adequate and complete interpretation of these data. These major flaws significantly affect the results and the authors' assessment.

1. The authors clearly state the aim of the study and detail the hypothesis they wish to test. However, it is very difficult for the reader to assess whether the setting selected is suitable to test their stated hypothesis. The authors give few details on the clinical relevance. The authors need to give details of patient characteristics. Did any individuals decline entry to the study? If so, how many patients declined to enter the study and which exclusion criteria were applied?

2. The present study is quite descriptive in its nature and does not sufficiently analyze the mechanisms, which lead to the role for Kupffer cells in coordinating cancer immunosurveillance and shaping outcomes to immunotherapy. The authors should further elaborate this question by additional experiments.

3. The authors should indicate the reason for using only one selective beta-glucan agent. This is a major disadvantage of the study design and makes the interpretation of data highly doubtful, especially since all the results of the study rely on these data. For validation purposes, additional experiments should be performed using an additional agent.

4. For experimental studies, several statistical methods exist to determine appropriate sample sizes, i.e. based on estimates of effect sizes (odds ratios). Did the authors attempt a sample size calculation? Otherwise how did the authors arrive at the number of samples used?

Reviewer #3 (expert in liver Immunology and Kupffer cells):

This study involves the role of macrophages within the liver invaded by pancreatic cancer cells. The authors modify the macrophage phenotype with B-glucan and show they can reduce tumor metastasis in the liver. The conclusion they make is that targeting Kupffer cells with B-glucan is the mechanism. I have a number of concerns I raise below.

1) While the authors do a nice job of showing preliminary data that they can selectively deplete Kupffer cells in the Clec4fdt mouse, they never use it for critical experiments to show that it is the Kupffer cells that are responsible for the anti-tumor effect. Instead they make use of chlodronate liposomes which presumably not only work better to deplete macrophages but also work better to deplete the bone marrow derived macrophages. As such I do not think that the authors can conclude that it is the Kupffer cells that are the target of B-glucan. The authors need to show loss of anti-tumor efficacy of B-glucan in Clec4fdt mice.

2) There are a number of properties of Kupffer cells that are not mentioned but should be considered. First, to my knowledge, Kupffer cells do not move but sit inside the sinusoids within the liver. As such they will never enter the tumor environment. Therefore, I assume that whatever the Kupffer cells are doing it is via the release of cytokines or other factors that affects distant tumors whether in the liver or in other organs. Is this the case? Do the authors know whether it is release of soluble factors that is mediating this biology? The bone marrow derived macrophages will enter the tumors in all organs and so another possible explanation for B-glucan working is that it stimulate bone marrow derived macrophages within tumors in all organs. This needs to be resolved.

3) Similarly, the T cells will interact with bone marrow derived macrophages in the tumor and with activated Kupffer cells directly or via release of cytokines but again it is not clear that it is the direct interaction that is mediating the T cell effects.

4) Giving an immune stimulator including cytokines LPS etc etc has all been tried in cancer and

usually works with the caveat that there are significant immune exacerbations. This is often seen with PD-1 inhibition. What side effects does giving B-glucan together with a PD-1 inhibitor have? I suspect it might be very significant.

5) I have in my time studying Kupffer cells never heard of them being referred to as being in the parenchyma of the liver which suggests that they are in among the hepatocytes. I am unsure whether the authors mean to say this, but certainly from the images in their figures and all literature I have ever read they are in the sinusoids and not in the parenchyma. Again some clarity around this would be helpful.

RESPONSE TO REVIEWERS' COMMENTS

REVIEWER #1: *“Thomas et al. described their study showing that β -glucan triggered activation of liver Kupffer cells (KC), suppressed cancer cell proliferation, and invoked anti-metastatic T cell immunity in pancreatic cancer mouse liver metastasis models. They also presented their analysis of pre- and on-treatment biopsy tumor specimens from the clinical trial that tested BG (odetiglucan) in combination with anti-PD-1 antibody in a Phase 2 study included patients with advanced triple negative breast cancer (TNBC) or melanoma with metastatic disease. It is nice to see a study combining both mouse model and human specimen analyses. Using a mouse model with a genetic depletion of KC is also a strength of this manuscript. However, the role of Kupffer cells in mediating treatment response to BG would need more data and a better mouse model to establish.”*

Comment #1: *“In the beginning of the manuscript, authors showed that CEL depletion of macrophages produced a significant decrease in metastatic tumor burden in the liver. However, in the remaining study, the manuscript is focused on showing depletion of macrophages leads to the abolish of antitumor effect of BG. Although these two opposite effects of macrophage depletion are not conflicted as the manuscript hypothesizes that macrophages can be reprogrammed/repolarized into antitumor macrophages, the data showing the protumor activity of CEL depletion of macrophages may not add much to the main theme of the macrophages and should be considered to move to the supplemental data.”*

Reply: We appreciate the suggestion from the Reviewer and have moved the data on CEL depletion of macrophages to the supplemental materials (see **Supplemental Fig. 6**).

Comment #2: *“Depletion of either BMDM or KC would decrease the treatment effects on the liver metastatic burdens. BG is also effective in the lung metastasis model. Therefore, the mouse experiment results doesn't seem to be able to establish the role of KC in mediating the treatment effect of BG.”*

Reply: The Reviewer is correct that BG can have effects against both liver and lung metastases. We included data to show this in our manuscript to demonstrate that the anti-metastatic activity of BG is not restricted to liver metastases. Consistent with this, a recent report (*Nat Immunol* 2023 24:239-254) showed the effects of whole beta-glucan particle on lung metastases in mouse models. However, the effect of a systemically delivered beta glucan on metastasis in pancreatic cancer has not been previously investigated. Here, we focused our studies on liver metastasis, the most common site of metastasis, and specifically interrogated the role of liver resident macrophages (Kupffer cells) in anti-tumor activity observed with BG. In our revised manuscript, we have included new data showing that depletion of Kupffer cells in Clec4f-DTR mice using diphtheria toxin (DT) abrogates the anti-metastatic activity of BG against liver metastases (**Fig. 2**). In addition, we have clarified our rationale for showing the effects of BG on lung metastasis. Specifically, we now state:

Line 158-164: *“We considered the possibility that the anti-metastatic activity of BG may not be restricted to the liver. Regarding this, we found that BG produced a significant reduction in tumor burden in a model of PDAC lung metastasis (**Supplementary Fig. 9**). This finding is consistent with prior work showing that whole BG particles can suppress lung metastasis in mouse models [40]. However, the anti-metastatic activity of BG has not been previously reported in PDAC. Therefore, we focused our subsequent investigations on the anti-metastatic mechanisms of BG in the PDAC liver metastasis setting.”*

Comment #3: *“Liver metastasis in the splenic vessel injection mouse model is not formed spontaneously. Authors should consider using the pancreatic orthotopic mouse model and examining the spontaneous liver metastases to study the treatment effect of BG with or without KC depletion.”*

Reply: The studies proposed here by the Reviewer would address the impact of BG on the early stages of metastasis including (i) invasion of cancer cells into the adjacent tissue of a primary tumor, (ii) intravasation into the blood stream, and (iii) survival in the blood. However, we purposely chose not to study these stages of the metastatic cascade given that metastasis is an early event in pancreatic cancer. Further, our studies in **Fig. 1k,l** and **Supplementary Fig. 10** show that BG does not impact metastatic seeding in the liver. We have revised our manuscript to clarify our approach. Specifically, we now state:

Line 102-107: *“To examine cell-cell interactions between liver macrophages and tumor cells as they seed and colonize the liver, we next modeled liver metastasis via intraportal (iPo) injection of tumor cells (**Supplementary Fig. 1f**). We purposely chose this approach because metastasis is an early event in PDAC[3-5] and with this model, we could study the late stages of the metastatic cascade when disseminated tumors cells encounter a distant organ.”*

Comment #4: *“Figure 4e-l showed that macrophage depletion produced a significantly higher metastatic burden in the liver of BG treated mice. In this experiment, a control group should be added to compare liver metastasis burden in BG treated mice vs. liver metastasis burden in untreated, CEL depleted mice.”*

Reply: The Reviewer is referring to our studies now presented in **Fig. 2** and **Supplementary Fig. 12** that address the requirement of liver macrophages for anti-metastatic activity induced by BG against liver metastases. As suggested, we now show the data with comparison to untreated mice as a control. Specifically, the data demonstrate that BG triggers anti-metastatic activity in the liver (Ctrl vs BG) and that this response is dependent on macrophages (BG vs BG + CEL) (**Supplementary Fig. 12**). Further, we now show using the Clec4f^{DTR} model that BG triggers anti-metastatic activity in the liver (Ctrl vs BG) and that this response is dependent specifically on Kupffer cells (BG vs BG+DT) (**Fig. 2**). We have revised the main text (**Lines 179-198**) accordingly.

Comment #5: *“For pre- and on-treatment biopsy specimen analysis, the results on liver metastases should be compared with other metastatic lesions. How the tumor areas are quantified is not described. Microscopically measuring the tumor areas is not previously validated. Radiographic responses should be used to characterize the antitumor effect of BG.”*

Reply: The Reviewer is referring to our immunofluorescence studies performed on tissue biopsies collected from patients with melanoma and triple negative breast cancer treated with BG in combination with anti-PD1. These studies are presented in **Fig 7**. The goal of these studies was to assess the capacity of BG + anti-PD1 to remodel metastatic lesions in patients. Paired analyses were performed to evaluate changes pre- and on-treatment. A dedicated study for comparing responses between metastatic lesions within the same patient and between patients is outside the scope of this manuscript and not feasible given the sample size and single lesion biopsies that were performed. As suggested, we have revised our manuscript to provide further details on how tumor areas were quantified. Specifically, we now state:

Line 554-565: *“Multispectral images were captured on the Vectra 3 imaging system (Akoya). Spectral unmixing and image analyses were performed using Inform Tissue*

Finder software (Akoya). Tumor regions of interest (ROIs) were defined, and cell segmentation and phenotyping was performed on the images. Imaging data containing marker intensities for each segmented cell was then converted into .csv files and imported into Flowjo for immune phenotyping. For each marker, the number of cells per sample was determined and used to calculate (i) tumor content per ROI area, (ii) macrophage infiltration/activation index, and (iii) T cell infiltration/activation index. Macrophage activation/infiltration index was calculated by the equation $[(\text{total macrophages}/\text{total tumor cells}) * ((\text{CD80}^+ \text{ macrophages}/\text{total macrophages}) / (\text{CD206}^+ \text{ macrophages}/\text{total macrophages}))]$. T cell activation/infiltration index was calculated by the equation $[(\text{total T cells}/\text{total tumor cells}) * ((\text{Ki67}^+ \text{ T cells}/\text{total T cells}) + (\text{Granzyme B}^+ \text{ T cells}/\text{total T cells}))]$.”

The Reviewer also suggests including radiographic response data. To this end, we have included a patient characteristics table (**Supplemental Table S5**) which displays radiographic responses for the patients presented in **Fig 7**.

Comment #6: “Clinical trial information should be provided in more detail, including patient demographic and clinicopathologic information. Radiographic responses should also be described.”

Reply: The Reviewer is referring to studies presented in **Fig 7** which show immunological effects of BG plus anti-PD1 in patients with triple negative breast cancer (TNBC, n=4) and melanoma (MEL, n=5). As suggested, we have added **Supplemental Table S5** which displays patient demographic and clinicopathologic information for these patients. Included in this table is radiographic response data.

REVIEWER #2: “In the manuscript entitled “Kupffer cells coordinate anti-metastatic activity and immune resistance in the liver”, Stacy Thomas and colleagues study the importance of macrophage polarity in metastasis and identify Kupffer cells as a novel therapeutic target to reverse cancer immune resistance. The rationale for the study seems reasonable as it is currently not clear how macrophage polarization could be an approach to intervention in metastasis. They report that macrophages respond to β -glucan treatment by inhibiting liver metastases. β -Glucan induced activation of macrophages in the liver (Kupffer cells), suppressed cancer cell proliferation, and elicited productive anti-metastatic T-cell immunity in mouse models of pancreatic cancer. Although these results are quite interesting, the manuscript has numerous deficiencies that preclude an adequate and complete interpretation of these data. These major flaws significantly affect the results and the authors' assessment.”

Comment #1: “The authors clearly state the aim of the study and detail the hypothesis they wish to test. However, it is very difficult for the reader to assess whether the setting selected is suitable to test their stated hypothesis. The authors give few details on the clinical relevance. The authors need to give details of patient characteristics. Did any individuals decline entry to the study? If so, how many patients declined to enter the study and which exclusion criteria were applied?”

Reply: The Reviewer is referring to data presented in **Fig 7** where we study the impact of BG plus anti-PD1 in patients with triple negative breast cancer and advanced stage unresectable melanoma. In these analyses, each patient underwent a pre- and on-treatment biopsy of the same non-target lesion to assess the immunological effects of treatment. Paired analyses were performed on matched biopsies from the same patient. The goal of these studies was to assess the capacity of BG + anti-PD1 to remodel metastatic lesions in patients based on the preclinical data presented in **Figs 1-6**. As suggested, we have included a patient demographics table

(**Supplementary Table S5**) that gives details of patient characteristics and tumor response. In addition, we have revised the Methods (**Lines 516-545**) to clarify inclusion and exclusion criteria for this study.

Comment #2: *“The present study is quite descriptive in its nature and does not sufficiently analyze the mechanisms, which lead to the role for Kupffer cells in coordinating cancer immunosurveillance and shaping outcomes to immunotherapy. The authors should further elaborate this question by additional experiments.”*

Reply: The Reviewer is referring to mechanisms by which Kupffer cells mediate anti-metastatic activity triggered by a beta glucan agonist (BG). To address this, we performed additional studies. In **Fig 3**, we present data showing that BG triggers enrichment of genes associated with an interferon response in Kupffer cells (**Fig 3e,f**). This effect is observed within 48 hours after treatment. Consistent with this, Kupffer cells respond *in vivo* to BG by increasing the transcription of genes associated with antigen presentation, including MHC genes (**Fig 3g, h, i**). Notably, BG did not induce transcriptional changes in the expression of chemoattractants or other cytokines in Kupffer cells. We confirmed these findings, detected by single cell RNA sequencing, using flow cytometry, where class I MHC molecules were observed to increase on Kupffer cells in response to BG treatment *in vivo* (**Fig 3j**). Together, these data suggested that Kupffer cells may stimulate anti-tumor T cells. Consistent with this, we found that BG increased interactions in the liver between Kupffer cells and T cells but not between bone marrow derived macrophages and T cells (**Fig 4a-c**). Further, BG treatment caused contraction of metastatic lesions with formation of a “contraction zone” that surrounded lesions and was infiltrated by T cells. Formation of this zone was blocked by depleting Kupffer cells (**Fig 5m**) and the anti-metastatic effects of BG were abrogated by depleting either Kupffer cells (**Fig 2d, e, f**) or T cells (**Fig 5h, i, j**). We then tested the effects of BG on tumor associated macrophages. Here, we found that BG caused an accumulation of tumor-associated macrophages expressing NOS2 and that this effect was dependent on both Kupffer cells and T cells (**Fig 6k, l**). Thus, these data indicate that the anti-metastatic activity triggered by BG is associated with a shift in the biology of tumor-infiltrating macrophages. Taken together, the findings support a model in which BG induces Kupffer cells to promote productive T cell surveillance against liver metastases. To summarize this new data we added to **lines 289-291** in our revised manuscript:

“In total, these data support a mechanism by which BG acutely activates KCs and T cells leading to re-education of BMDMs to control liver metastasis.”

Comment #3: *“The authors should indicate the reason for using only one selective beta-glucan agent. This is a major disadvantage of the study design and makes the interpretation of data highly doubtful, especially since all the results of the study rely on these data. For validation purposes, additional experiments should be performed using an additional agent.”*

Reply: Our study focuses on the use of a systemically delivered soluble beta glucan agonist (BG, odetiglucan). We focused on this agent because it is currently in clinical development for patients with advanced cancer and has been administered to more than 500 cancer patients in a series of clinical trials and in combination with tumor targeting Abs, anti-angiogenic Abs, and with ICIs. (NCT00545545, NCT00912327, NCT00874107, NCT00874848, NCT01309126, NCT02981303, NCT00542217). Other beta glucans of different structure (e.g. soluble vs particulate) or from different organisms (e.g. source of beta-glucan) are not currently in clinical development, and thus, further understanding of the biological activity of this agent holds significant clinical relevance. Our studies were not performed with the intent to compare beta glucans, rather the

studies presented in our manuscript reveal important mechanistic insight into the anti-metastatic activity of BG in liver metastasis, a setting where immunotherapies have shown very little clinical efficacy. Our studies show that odetiglucan has anti-metastatic activity that is reliant on Kupffer cells which are necessary for T cell-dependent anti-tumor activity triggered by BG. Preclinical studies also reveal that anti-PD1 can enhance the anti-metastatic activity of BG and translational studies using tissue samples from patients treated with BG plus anti-PD1 support the mechanism of action observed preclinically. Overall, our study provides new and translationally significant findings and offers insights into mechanisms regulating immune surveillance against liver metastases. We have clarified the premise of our studies in our revised manuscript. Specifically, we now state:

Lines 82-83: “Here, we studied a soluble β -1,3/1,6 glucan (odetiglucan) that is already in clinical development.”

Lines 152-153: “Odetiglucan is currently in clinical development and is a novel Dectin-1 (CLEC7A) agonist.”

Comment #4: “For experimental studies, several statistical methods exist to determine appropriate sample sizes, i.e. based on estimates of effect sizes (odds ratios). Did the authors attempt a sample size calculation? Otherwise how did the authors arrive at the number of samples used?”

Reply: Sample sizes were estimated based on pilot experiments conducted in the laboratory and were selected to provide sufficient numbers of mice in each group to yield a two-sided statistical test, with the potential to reject the null hypothesis with a power ($1-\beta$) of 80%, subject to $\alpha = 0.05$. We have revised the Methods on **Lines 625-628** to include this information.

REVIEWER #3: “This study involves the role of macrophages within the liver invaded by pancreatic cancer cells. The authors modify the macrophage phenotype with B-glucan and show they can reduce tumor metastasis in the liver. The conclusion they make is that targeting Kupffer cells with B-glucan is the mechanism. I have a number of concerns I raise below.”

Comment #1: “While the authors do a nice job of showing preliminary data that they can selectively deplete Kupffer cells in the *Clec4fdt* mouse, they never use it for critical experiments to show that it is the Kupffer cells that are responsible for the anti-tumor effect. Instead, they make use of chlodronate liposomes which presumably not only work better to deplete macrophages but also work better to deplete the bone marrow derived macrophages. As such I do not think that the authors can conclude that it is the Kupffer cells that are the target of B-glucan. The authors need to show loss of anti-tumor efficacy of B-glucan in *Clec4fdr* mice.”

Reply: In our revised manuscript, we now show that depletion of Kupffer cells in *Clec4f-dtr* mice blocks the anti-metastatic activity of BG (**Fig 2d, e, f**). Thus, the data demonstrate that Kupffer cells are required for BG-induced anti-metastatic activity against liver metastases.

Comment #2: “There are a number of properties of Kupffer cells that are not mentioned but should be considered. First, to my knowledge, Kupffer cells do not move but sit inside the sinusoids within the liver. As such they will never enter the tumor environment. Therefore, I assume that whatever the Kupffer cells are doing it is via the release of cytokines or other factors that affects distant tumors whether in the liver or in other organs. Is this the case? Do the authors know whether it is release of soluble factors that is mediating this biology? The bone marrow

derived macrophages will enter the tumors in all organs and so another possible explanation for B-glucan working is that it stimulate bone marrow derived macrophages within tumors in all organs. This needs to be resolved.”

Reply: The Reviewer is referring to mechanisms by which Kupffer cells mediate anti-metastatic activity triggered by a beta glucan agonist (BG). To address this, we performed additional studies. In **Fig 3**, we present data showing that BG triggers enrichment of genes associated with an interferon response in Kupffer cells (**Fig 3e,f**). This effect is observed within 48 hours after treatment. Consistent with this, Kupffer cells respond *in vivo* to BG by increasing the transcription of genes associated with antigen presentation, including MHC genes (**Fig 3g, h, i**). Notably, BG did not induce transcriptional changes in the expression of chemoattractants or other cytokines in Kupffer cells. We confirmed these findings, detected by single cell RNA sequencing, using flow cytometry, where class I MHC molecules were observed to increase on Kupffer cells in response to BG treatment *in vivo* (**Fig 3j**). Together, these data suggested that Kupffer cells may stimulate anti-tumor T cells. Consistent with this, we found that BG increased interactions in the liver between Kupffer cells and T cells but not between bone marrow derived macrophages and T cells (**Fig 4a-c**). Further, BG treatment caused contraction of metastatic lesions with formation of a “contraction zone” that surrounded lesions and was infiltrated by T cells. Formation of this zone was blocked by depleting Kupffer cells (**Fig 5m**) and the anti-metastatic effects of BG were abrogated by depleting either Kupffer cells (**Fig 2d, e, f**) or T cells (**Fig 5h, i, j**). We then tested the effects of BG on tumor associated macrophages. Here, we found that BG caused an accumulation of tumor-associated macrophages expressing NOS2 and that this effect was dependent on both Kupffer cells and T cells (**Fig 6k, l**). Thus, these data indicate that the anti-metastatic activity triggered by BG is associated with a shift in the biology of tumor-infiltrating macrophages. Taken together, the findings support a model in which BG induces Kupffer cells to promote productive T cell surveillance against liver metastases. To summarize this new data we added to **lines 289-291** in our revised manuscript:

“In total, these data support a mechanism by which BG acutely activates KCs and T cells leading to re-education of BMDMs to control liver metastasis.

Comment #3: “*Similarly, the T cells will interact with bone marrow derived macrophages in the tumor and with activated Kupffer cells directly or via release of cytokines but again it is not clear that it is the direct interaction that is mediating the T cell effects.*”

Reply: We address this comment above under **Comment #2** where we provide new data showing that Kupffer cells upregulate molecules involved in antigen presentation and show increased interaction between Kupffer cells and T cells in response to BG. These findings are consistent with a prior report showing that Kupffer cells can cross-present hepatocellular antigens to improve the antiviral function of T cells (*Immunity* 2021 54:2089-2100.e8). We have revised our manuscript to incorporate a discussion on these new findings. Specifically, we now state:

Lines 347-361: “ β -glucan treatment stimulated Kupffer cells to coordinate anti-metastatic activity that was dependent on T cells. We found that treatment caused an immune reaction at the border of metastatic lesions which we termed a “contraction zone”. Both Kupffer cells and T cells were required for formation of this immune reaction which was notably devoid of tumor cells. In addition, T cells were found within the contraction zone, but Kupffer cells were excluded. In contrast, we detected T cell-Kupffer cell interactions in the liver that were increased by β -glucan treatment. Together, these findings raise the possibility that Kupffer cells may stimulate tumor-specific T cell responses through antigen presentation which occurs outside of metastatic lesions but within the liver. Consistent

with this, we found that β -glucan treatment increased Kupffer cell expression of *Irf7*, which is known for its critical role in interferon responses that govern the induction of T cell immunity [48]. β -glucan treatment also increased expression of *Isg15* in Kupffer cells. *Isg15* is an interferon-induced ubiquitin-like protein with immunomodulatory potential capable of enhancing APC function [49]. Thus, these findings support a role for polarizing Kupffer cells to invoke anti-metastatic responses and emphasize the importance of bridging innate and adaptive immunity for cancer immunotherapy.”

Comment #4: *“Giving an immune stimulator including cytokines LPS etc etc has all been tried in cancer and usually works with the caveat that there are significant immune exacerbations. This is often seen with PD-1 inhibition. What side effects does giving B-glucan together with a PD-1 inhibitor have? I suspect it might be very significant.”*

Reply: The Reviewer is referring to the safety profile of BG when administered in combination with anti-PD1. Specifically, in the IMPRIME 1 study, the combination was overall well tolerated in advanced melanoma and TNBC populations and the safety profile was as expected based on underlying disease and prior experience with BG and anti-PD1. The most commonly reported events of Grade 1 or 2 infusion reactions were generally manageable with treatment or study drug interruption.

Comment #5: *“I have in my time studying Kupffer cells never heard of them being referred to as being in the parenchyma of the liver which suggests that they are in among the hepatocytes. I am unsure whether the authors mean to say this, but certainly from the images in their figures and all literature I have ever read they are in the sinusoids and not in the parenchyma. Again some clarity around this would be helpful.”*

Reply: The Reviewer is correct, and we have revised our manuscript accordingly to clarify the location of Kupffer cells within the sinusoids of the liver. Specifically, we now state:

Lines 118-120: “Further, in established metastases (outgrowth phase), BMDMs largely existed within lesions. In contrast and as expected, KCs, which reside in the liver sinusoids, were restricted to the adjacent liver tissue.”

REVIEWERS' COMMENTS

Reviewer #1 (Remarks to the Author):

The revised manuscript has addressed many of my prior comments and suggestions. However, one of the key issues about the mouse model was not addressed. Reading Reviewer #3's comments about the location of Kupffer cells, I think it is critical and necessary to repeat the experiment in the pancreatic orthotopic mouse model to examine the spontaneous formation of liver metastasis. It has been the same puzzle to me how depletion of Kupffer cells would lead to author's observations in the splenic injection model where liver metastases are implanted directly and formed diffusely. Nevertheless, it is possible that Kupffer cells may modulate the niches that host the spontaneous liver metastasis formation in the setting of BG treatment. The author group has used the orthotopic model in their prior publications and should be able to incorporate the orthotopic model to this study.

Reviewer #2 (Remarks to the Author):

In the manuscript entitled "Kupffer cells coordinate anti-metastatic activity and immune resistance in the liver", Stacy Thomas and colleagues study the importance of macrophage polarity in metastasis and identify Kupffer cells as a novel therapeutic target to reverse cancer immune resistance.

Although this article is very interesting, there were a few deficiencies within the manuscript, which precluded adequate and full interpretation of these data. The authors addressed all comments raised by this reviewer and the article appears majorly improved. As requested, they added patient characteristics and tumor response. In addition, they have revised the methods to clarify inclusion and exclusion criteria for this study. They also adapted figure 3 to support the mechanism by which BG acutely activates KCs and T cells leading to re-education of BMDMs to control liver metastasis. Lastly, they estimated sample sizes based on pilot experiments conducted in the laboratory and revised the methods to include this information.

Reviewer #2 was also asked to arbitrate on the responses to Reviewer #1:

I carefully reviewed the manuscript and the authors' point-by-point response to reviewer #1.

The reviewer believes that the role of Kupffer cells in mediating treatment response to odetiglucon needs a better mouse model to establish.

Therefore, the reviewer asks the authors to consider using an orthotopic pancreas mouse model to study the spontaneous development of liver metastases in order to investigate the treatment effect of odetiglucon with or without Kupffer cell depletion.

In my opinion, the model proposed by the reviewer would focus on the effect of odetiglucon on the early stages of metastasis.

These stages of the metastatic cascade do not necessarily need to be studied since metastasis is an early event in pancreatic cancer.

I would not necessarily insist on the implementation of this mouse model.

For this reason, in my opinion, it can be currently dispensed with.

Reviewer #3:

This reviewer was not available for a second review. Reviewer #4 was asked to assess the authors' responses to this reviewer.

Reviewer #4 (Remarks to the Author):

As specifically requested, I reviewed the concerns raised by reviewer 3 and the answers provided by the authors. The authors addressed the comments of reviewer 3 correctly, and provided

additional data supporting their claim on the role of KCs in metastatic outgrowth and BG efficacy.

RESPONSE TO REVIEWERS' COMMENTS

Editor:

Comment 1: *Please revise your manuscript files as follows:*

1. *Please submit the article file as a word (docx.) file. Any changes made should be tracked using Microsoft Word's "tracked changes" feature.*

Reply: A *.docx file has been prepared including a "tracked changes" and "clean" version.

2. *I have started editing your manuscript abstract and title. Please use the attached Word (docx.) file to make any further changes. Any changes made should be tracked using Microsoft Word's "tracked changes" feature. There is also a comment from me in that file that requires your attention.*

Reply: The title has been updated as suggested to, "Kupffer cells prevent pancreatic ductal adenocarcinoma metastasis to the liver in mice." The abstract has been revised to clarify the text, "remodeled the immune response" to " β -glucan drove T cell activation and macrophage re-polarization in liver metastases in mice and humans and sensitized metastatic lesions to anti-PD1 therapy." as in **line 40**.

3. *Please check that the order of the subsections of the text meets the requirements of our journal (see table below for instructions).*

Reply: The order of subsections has been reviewed and matches the requirements of *Nature Communications*.

4. *Please submit the figures as individual files. Figure legends should be removed from the figures themselves. The labels 'Figure 1, Figure 2 etc.' should be removed from the figures themselves.*

Reply: Individual figure files are now provided with figure legends and labels removed as suggested.

Comment 2: *Please note we do not allow the use of phrases referring to the findings as being ‘new, novel or the first’ (even if to the best of your knowledge, they are indeed new, novel or the first). Please check thoroughly, as we may experience delays in publication if the text is not fully edited to meet this requirement.*

Reply: All references to “new, novel or the first’ have been removed.

Comment 3: *In the source data file, please provide the source data for all Kaplan-Meier curves, as this is currently missing.*

Reply: Source data for Kaplan-Meier curves (**Fig. 1p** and **Fig. 7f**) are included in a revised source data file.

Comment 4: *Please ensure all sequencing/microarray/proteomics datasets are released from embargo, as we are unable publish your manuscript until they are made available for public access. In the Data availability section, please use this format: “The [DATA] have been deposited in [DATABASE] accession code [insert code] [insert hyperlink to datasets]”. Please refer to our site (<https://www.nature.com/nature-portfolio/editorial-policies/reporting-standards>) for more information on reporting standards. The manuscript will be returned to you without processing if the data remain private.*

Reply: All data are set for public release on September 4, 2023. See **line 651** in the Data Availability Section.

Comment 5: *Please check your Code availability statement, as this suggests that custom was used but not disclosed at peer review. In addition, there is no mention of custom code in the methods.*

Please clarify how this code was used and where. Please also clarify whether and how this will be made available to other researchers after publication.

Reply: To clarify the Code availability, we have revised our text to state in **Line 656 – 657:**

“No custom algorithms were used. All code used to create figures is available in R packages as detailed in the Methods.”

Comment 6: *Please describe your sources of funding in the Acknowledgements section.*

Reply: A statement of **Funding** was added in the revised manuscript as in **Line 865 – 872.**

Comment 7: *Please address the following changes regarding the methods:*

a. Please state the full name of the IRB that approved the use of human samples, as ‘New England IRB’ does not seem to match any of the authors’ affiliations.

Reply: The referenced study from which human samples were collected was conducted by the Pancreatic Cancer Action Network (PanCAN). De-identified samples were then provided to us from PanCAN as part of a grant mechanism. As such, the IRB referenced is correct. To clarify this, we have revised the Methods on **line 513** to indicate “Biopsy samples from human PDAC liver metastasis lesions, collected from patients enrolled in the Pancreatic Cancer Action Network (PanCAN) Know Your Tumor (KYT) program[77], were provided by PanCAN.

b. *Please provide an explicit statement in the related Methods section indicating that the use of human material from the IMPRIME-1 trial has been approved by your institutional ethics review committee. The name of the committee (and approval number where applicable) should also be stated.*

Reply: The central and local IRBs approving the IMPRIME-1 study were added to the methods in **Line 526 – 529.**

c. *Please also confirm whether all recruited volunteers from that trial provided written informed consent. This should also be stated in the Materials and methods section. Otherwise, if your ethics review board has waived the requirement for consent from the patients, please let me know (please attach a copy of the waiver/approval letter in the next submission).*

Reply: Written consent was obtained from all patients or their guardians. In our revised manuscript this is now stated in **Line 525 – 526.**

d. *In the Methods section, please add an explicit statement specifically confirming that the protocols used for animal experimentation were approved by an animal ethics/welfare committee. The name of the committee (and approval number where applicable) should also be stated.*

Reply: The **Animal Study** section of the Methods in our revised manuscript has been updated to confirm the protocol and protocol # used for animal experimentation as in **Lines 446 – 447.**

e. *Please provide the stock# for all purchased animal strains in the Methods.*

Reply: The **Animal Study** section of the Methods in our revised manuscript has been updated to identify the stock# of all mice used as in **Lines 441 -443**.

f. *In the Methods, for non-commercial animal strains, please provide reference(s) unless this is a new mouse model, which then needs to be described in full details.*

Reply: Only commercial strains were used. Breeding of Clec4f-cre^{+/-} Rosa-DTR^{flox/+} is described in **Lines 443 – 444**.

g. *In the Methods section, please state the type of animal facility (barrier, specific pathogen-free/SPF, or pathogen-free/germ-free/GF), and whether experimental/control animals were co-housed or bred separately.*

Reply: Pathogen-free barrier animal facility was used as stated in **Line 444 – 445**.

h. *In the Methods section, please state the method for euthanasia, and avoid stating ‘sacrifice’ when applicable.*

Reply: CO2 inhalation was used as in **Line 447 – 448**.

i. *Please ensure the number, sex, strain and age of mice is clearly stated for all experiments, either in the Methods section or in the respective figure legends.*

Reply: The Methods and figure legends have been revised to specify the sex, strain and age of mice for all experiments.

j. *In the Methods sections, please state the maximum tumour size allowed by your institutional ethics board and confirm that you have adhered to these size limits in your experiments. Please also include a statement outlining how you measured tumour size and how frequently these animals were monitored, as well as the humane endpoints used to determine when the experiments should be terminated to prevent animal suffering.*

Reply: The Methods (**lines 448 – 451**) now state, “Mice were monitored three times per week and euthanized based on defined criteria including loss of > 20% body weight, body condition score > 2, lethargy or other signs of distress. Metastatic liver lesions were not grossly measurable and thus tumor measurements were not routinely taken.”

Comment 8: Please make the following changes to the figures:

a. *Please ensure all graphs have error bars and that data presentation is described in the legend (e.g. mean and SD or SEM, median and range etc.).*

Reply: All graphs were reviewed and confirmed to display mean \pm SD, which is described in the associated figure legends.

b. *Please increase the size of all microscopy images, as they are currently much too small to distinguish any histological features or specific antigen staining. Any high-magnification inset should also have a scale bar (this is missing in some).*

Reply: The figures were updated with larger microscopy images and to add scale bars where needed.

c. *In Fig. 4a, please remove the white background of the labels 1,2,3 and 4 as this is obscuring the micrograph itself.*

Reply: The white background of labels in **Figure 4a** were removed.

d. *In Fig. 5c, please add a scale bar to the right-hand panel.*

Reply: A scale bar was added to the right-hand panel of **Fig. 5c**.

e. *In Fig. 5d and e, please ensure the symbol colours match the group names. I appreciate that orange was chosen because both groups come from BG treatments, but the figure would be easier to read if the groups matched the anatomical locations (tumour vs contraction in blue and green respectively).*

Reply: **Fig. 5d** and **5e** were updated to match the symbol colors with the group names.

f. In Fig. 6k, what is the area shown in the high-magnification inset? Please show this as a square in the lower-magnification image.

Reply: The images in **Fig. 6k** were revised to include a square highlighting the region chosen for high-magnification.

g. In Figs. 7g-k. The high-magnification inset is particularly poor and pixelated or blurry.. Please provide an image of improved quality and resolution. Please also add scale bars.

Reply: New images were inserted into **Figs 7g-k** at a higher resolution and scale bars were added.

Comment 9: *Please make the following changes to the supplementary figures*

a. The first page of the supplementary information file should contain the title of the study and author list (similar to the main text file).

Reply: A title page was added to the supplementary file.

b. Please increase the size of all microscopy images, as they are currently much too small to distinguish any histological features or specific antigen staining. Any high-magnification inset should also have a scale bar (this is missing in some).

Reply: The supplementary figures were edited to include larger microscopy images and add scale bars where needed.

c. In supplementary Fig. 1b, 1d 6c, please ensure the labels I and ii do not have any white background as this is obscuring a large part of the high-magnification insets.

Reply: The white background was removed from **supplementary figures 1b, 1d and 6c**.

5. **Comment 10:** *Lastly, please ensure that all points raised in the reporting summary have been addressed. You can visualise all comments made by hovering over the small yellow comment bubbles on the left-hand side.*

Reply: All queries raised in the reporting summary have been addressed in our revised manuscript.

Reviewer 1: *The revised manuscript has addressed many of my prior comments and suggestions. However, one of the key issues about the mouse model was not addressed. Reading Reviewer #3's comments about the location of Kupffer cells, I think it is critical and necessary to repeat the experiment in the pancreatic orthotopic mouse model to examine the spontaneous formation of liver metastasis. It has been the same puzzle to me how depletion of Kupffer cells would lead to author's observations in the splenic injection model where liver metastases are implanted directly and formed diffusely. Nevertheless, it is possible that Kupffer cells may modulate the niches that host the spontaneous liver metastasis formation in the setting of BG treatment. The author group has used the orthotopic model in their prior publications and should be able to incorporate the orthotopic model to this study.*

Reply: We thank the reviewer for their careful appraisal of our manuscript and their suggestion to conduct experiments in the orthotopic setting. However, as discussed in our revised manuscript, studies proposed here by the Reviewer would address the impact of BG on the early stages of metastasis including (i) invasion of cancer cells into the adjacent tissue of a primary tumor, (ii) intravasation into the blood stream, and (iii) survival in the blood. However, we purposely chose not to study these stages of the metastatic cascade given that metastasis is an early event in pancreatic cancer. Further, our studies in **Fig. 1k,l** and **Supplementary Fig. 10** show that BG does

not impact metastatic seeding in the liver. We have revised our manuscript to clarify our approach. Specifically, in our revised manuscript we state:

Line **103-108**: “To examine cell-cell interactions between liver macrophages and tumor cells as they seed and colonize the liver, we next modeled liver metastasis via intraportal (iPo) injection of tumor cells (**Supplementary Fig. 1f**). We purposely chose this approach because metastasis is an early event in PDAC[3-5] and with this model, we could study the late stages of the metastatic cascade when disseminated tumors cells encounter a distant organ.”

Reviewer 2: *In the manuscript entitled “Kupffer cells coordinate anti-metastatic activity and immune resistance in the liver”, Stacy Thomas and colleagues study the importance of macrophage polarity in metastasis and identify Kupffer cells as a novel therapeutic target to reverse cancer immune resistance.*

Although this article is very interesting, there were a few deficiencies within the manuscript, which precluded adequate and full interpretation of these data. The authors addressed all comments raised by this reviewer and the article appears majorly improved. As requested, they added patient characteristics and tumor response. In addition, they have revised the methods to clarify inclusion and exclusion criteria for this study. They also adapted figure 3 to support the mechanism by which BG acutely activates KCs and T cells leading to re-education of BMDMs to control liver metastasis. Lastly, they estimated sample sizes based on pilot experiments conducted in the laboratory and revised the methods to include this information.

*Reviewer #2 was also asked to arbitrate on the responses to Reviewer #1:
I carefully reviewed the manuscript and the authors' point-by-point response to reviewer #1.*

The reviewer believes that the role of Kupffer cells in mediating treatment response to Odetiglucan needs a better mouse model to establish. Therefore, the reviewer asks the authors to consider using an orthotopic pancreas mouse model to study the spontaneous development of liver metastases in order to investigate the treatment effect of odetiglucan with or without Kupffer cell depletion.

In my opinion, the model proposed by the reviewer would focus on the effect of odetiglucan on the early stages of metastasis. These stages of the metastatic cascade do not necessarily need to be studied since metastasis is an early event in pancreatic cancer. I would not necessarily insist on the implementation of this mouse model. For this reason, in my opinion, it can be currently dispensed with.

Reply: We thank the reviewer for their comments and suggestions which have greatly improved the content of this manuscript.

Reviewer 3: not available

Reviewer 4: *As specifically requested, I reviewed the concerns raised by reviewer 3 and the answers provided by the authors. The authors addressed the comments of reviewer 3 correctly,*

and provided additional data supporting their claim on the role of KCs in metastatic outgrowth and BG efficacy.

Reply: We greatly appreciate the Reviewer's additional evaluation of our work.